# CONSISTENT DIFFUSION LANGUAGE MODELS

## ABSTRACT

Diffusion-based language models (DLMs) have emerged as compelling alternatives to sequential autoregressive generation, offering the promise of parallel decoding. Yet existing discrete diffusion models require hundreds of refinement steps for high-quality text, undermining the efficiency gains of parallelism. We introduce the Consistent Diffusion Language Model (CDLM), a new family of generative models that brings the benefits of consistency training—enforcing agreement across noise levels to enable one- or few-step generation—to the discrete domain. Our approach leverages an exact closed-form formulation of discrete posteriors, providing a rigorous analogue to the missing probability-flow ODE in discrete space. This yields a multi-path consistency objective that, as we show, unifies and generalizes popular diffusion, consistency, and distillation methods in a single view. To ensure stability at scale, we introduce a set of principled design choices that prevent training pathologies like mode collapse. On conditional and unconditional text-generation benchmarks, CDLM establishes new state of the art as a single-stage model, consistently outperforming both base and distilled DLMs across sampling budgets. These results position CDLM as a new paradigm for efficient, scalable, and high-fidelity discrete generative modeling.

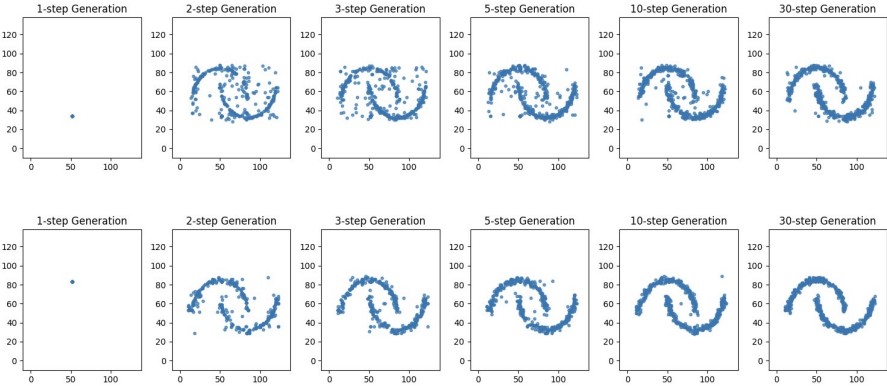

Figure 1: **Illustrative toy example on 2D moons under discrete diffusion.** The continuous moons data are quantized into tokens and modeled as a language-like sequence. Standard diffusion language model (top) only forms a sharp structure after 10+ denoising steps, while CDLM (bottom) not only yields clear samples within 2–3 steps, but also continues to improve with larger budgets.

## 1 INTRODUCTION

Diffusion models have emerged as a dominant paradigm in generative modeling, achieving state-of-the-art results in image, audio, and video generation (Yang et al., 2023). Their appeal lies in a simple principle of iterative refinement: data are gradually corrupted into noise and then reconstructed step by step, with each denoising step refining the sample toward the data distribution. This iterative view has proven both scalable and versatile across continuous domains.

Recently, the diffusion paradigm has extended to language, where its promise lies not just in quality but perhaps moreso in efficiency (Austin et al., 2021a). Unlike autoregressive (AR) models

that are constrained to sequentially decode token-by-token, diffusion language models can generate and refine multiple tokens in parallel, suggesting a path toward sublinear-time text generation and flexible, controllable synthesis (Li et al., 2022). Among them, masked diffusion language models (MDLMs), where corruption is defined by masking tokens, have shown strong empirical results, even rivaling autoregressive baselines more recently (Sahoo et al., 2024; Shi et al., 2024; Nie et al., 2025a). However, the potential of MDLMs remains largely untapped in practice, since high-quality generation typically requires hundreds of refinement steps, which erodes the computational gains of parallelism. Speeding up these models has become a central open challenge for discrete diffusion.

In continuous domains, transformative acceleration techniques like consistency models (Song et al., 2023) and progressive distillation (Salimans & Ho, 2022) have helped meet the promise of diffusion by enabling effective few- or even one-step generation. These approaches critically rely on the existence of a probability flow ordinary differential equation (PF-ODE), which defines a unique, deterministic trajectory from any noisy point $x_t$ back to the data $x_0$. In discrete space, however, no such ODE exists, meaning there is no single path that ties all noise levels together. This absence has so far prevented discrete diffusion models from benefiting from similar acceleration frameworks.

In this paper, we introduce a new principle for discrete generative modeling: *multi-path consistency*. Rather than searching for a non-existent unique trajectory, we effectively embrace the multiplicity of possible denoising paths in discrete space. Specifically, any two noise levels $s < t$ can be connected by an exact posterior bridge, which we show is available in closed form for common corruptions including masking. These bridges define a rich family of stochastic paths, from a single direct jump to a chain of many small steps, all of which ultimately reconstruct the same clean data. Our central insight is that by training a model to be consistent across these paths, we can learn a predictor whose output is *invariant* to the denoising path taken. And when long paths (many small steps) and short paths (few large steps) are trained to yield the same prediction, we can choose to benefit from the more efficient, shorter path. Few-step generation emerges as a "consequence" of path-equivalence enforced through multi-path consistency.

Building on this principle, we propose the *Consistent Diffusion Language Model (CDLM)*, a new family of discrete generative models that turns few-step efficiency into a *training-time property*. CDLM trains a time-conditional predictor $f_\theta(x_t, t)$ by enforcing agreement across pairs $(x_t, t)$ and $(x_s, s)$, where $x_s$ is sampled from the exact posterior bridge $q(x_s \mid x_t, x_0)$. In effect, the model is asked to make its prediction at the noisier state $x_t$ consistent with its prediction at the cleaner state $x_s$. This consistency enforces an implicit decomposition of the denoising task: predicting from $x_t$ is equivalent to first "hopping" to $x_s$ via the true bridge, and then solving the simpler denoising step from $x_s$ toward $x_0$. By enforcing this consistency across many such bridges and step sizes, CDLM acquires the ability to traverse different routes from corrupted to clean inputs, learning not only gradual multi-step refinements but also direct, long-range transitions. This property allows CDLM to generate high-quality outputs in just a handful of steps, while still improving the generation with more steps rather than saturating. We compare our models with base models that are trained from scratch within a single stage including MDLM (Sahoo et al. (2024)) and DUO (Sahoo et al. (2025)) , as well as distilled models that performs an additionally dedicated stage of distillation on the base models, including SDTT (Deschenaux & Gulcehre (2025)) and DUO+DCD (Sahoo et al. (2025)). As shown in Figure 2, Table 1 and 4, our model delivers strong empirical results that consistently outperformed state-of-the-art base and distilled diffusion models at the same scale for text generation, regardless of sampling steps, precision and conditions.

Our contributions are threefold:

1. *A new principle for discrete generative modeling.* We introduce multi-path consistency and show how to enforce it using exact posterior bridges as a powerful substitute for the absent PF-ODE trajectory, providing a rigorous and general foundation for efficient and effective discrete generation.

2. *A unified, generalizable, and robust training framework.* We present a single, self-contained objective for training a new class of consistency Diffusion Language Models to be invariant across denoising paths. We further develop novel connections with popular generative modeling techniques, showing that standard diffusion, consistency, and distillation-like behaviors emerge as special cases of CDLM. We complement it with a suite of practical techniques that ensure stable and scalable training, and show the effectiveness for both Masked and Uniform diffusion prior.

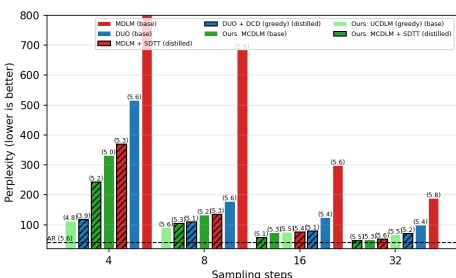 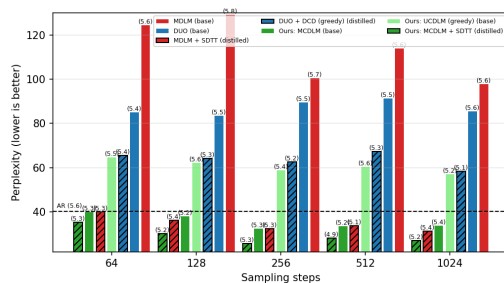

Figure 2: Perplexity (entropy) vs. sampling steps with 64-bit sampler for unconditional generation. Base models are without edges and hatches, while distilled models are indicated by shadow hatched bars ▨. We use Red for MDLM based models, Blue for DUO based models, and Green for our CDLM based models (MCDLM and UCDLM denotes model with *Masked* and *Uniform* prior). We pick the best two models for each family, while including more details on Section 4.3.1. For base MCDLM model, we chose MCDLM-PPLOptimized, a variant trained to achieve much better perplexity with slightly lower entropy, which outperforms all other base models for all sampling steps, and also beats distilled models under a majority of the steps while maintaining a similar entropy. Likewise, our distilled MCDLM further delivers best performance among distilled models with similar diversity. Note that DUO+DCD with greedy sampler has a significantly lower entropy (3.9) which often indicates poor sampling diversity and a biased perplexity.

3. *State-of-the-art text generation.* On standard conditional and unconditional text generation benchmarks, CDLM, as a single stage model, establishes new SOTA across varied sampling budgets regardless of the sampling precision, outperforming both base DLMs as well as distilled DLMs that are optimized at a second stage specifically for generation perplexity. A distilled version of CDLM is further introduced to achieve even better generation while maintaining the diversity. Together the models also achieve a 16-32x speedup for 32 and 64-bit sampling comparing with autoregressive baseline.

CDLM reframes discrete diffusion as training a *path-independent denoiser*. By enforcing consistency across many stochastic bridges, the model learns to map corrupted inputs to clean text in a manner that is efficient and robust, and scalable. This unifies diffusion and consistency perspectives, and establishes multi-path consistency as a new paradigm for discrete generative modeling.

## 2  PROBLEM SETUP

We ground our framework in the standard formalism of discrete diffusion (Austin et al., 2021a), which defines a forward-time corruption process that gradually transforms data into noise, together with a parameterized reverse process that reconstructs data from noise. In discrete domains such as text, states are sequences of categorical variables. To avoid ambiguity between sequence-level and token-level operations, let $\boldsymbol{x} \in \mathcal{V}^L$ denote a sequence of length $L$ over a vocabulary $\mathcal{V}$. The state at time $t$, $\boldsymbol{x}_t = (x_t^1, \ldots, x_t^L)$, consists of discrete tokens $x_t^i \in \mathcal{V}$. The forward process is a non-homogeneous Markov chain that factorizes independently over token positions. For a single token position, the transition is governed by matrices $\boldsymbol{Q}_t \in \mathbb{R}^{|\mathcal{V}| \times |\mathcal{V}|}$:

$$q(\boldsymbol{x}_t \mid \boldsymbol{x}_0) = \prod_{i=1}^{L} q(x_t^i \mid x_0^i) = \prod_{i=1}^{L} \mathrm{Cat}(x_t^i;\ x_0^i \boldsymbol{Q}_{1:t}), \quad \text{with} \quad \boldsymbol{Q}_{1:t} = \prod_{s=1}^{t} \boldsymbol{Q}_s. \tag{1}$$

where $\boldsymbol{x}_0^i$ denotes the one-hot vector of the token at position $i$. Each $\boldsymbol{Q}_t$ is row-stochastic to conserve probability mass. Additionally, rows of $\boldsymbol{Q}_{1:t}$ must converge to a known stationary distribution over time, ensuring that $q(\boldsymbol{x}_t)$ approaches a tractable prior over time. Using $\langle \cdot, \cdot \rangle$ for Euclidean inner product and $\odot$ for elementwise product, the exact posterior at time $t-1$ for a single token $x^i$ is written in closed form:

$$q(x_{t-1}^i \mid x_t^i, x_0^i) = \mathrm{Cat}\left(x_{t-1}^i;\ \frac{x_t^i \boldsymbol{Q}_t^\top \odot x_0^i \boldsymbol{Q}_{1:t-1}}{\langle x_0^i \boldsymbol{Q}_{1:t}, x_t^i \rangle}\right). \tag{2}$$

A particularly important instance for language is *masked (or absorbing-state) diffusion*, where the stationary distribution places all probability on a special `[MASK]` token. Not only is masking found

to be the most effective corruption (Austin et al., 2021a; Sahoo et al., 2024; Shi et al., 2024; Nie et al., 2025a), but it also allows helps simplify the closed-form marginals and posteriors. Masked diffusion language models exploit this to parameterize $\boldsymbol{x}_0$ directly, enabling efficient, parallel sampling.

**Continuous diffusion and a natural notion of consistency.** In continuous domains, a standard construction of the forward process is often expressed as a stochastic differential equation (SDE):

$$d\boldsymbol{x}_t = g(t)\, d\boldsymbol{w}_t, \qquad t \in [0,1], \quad \boldsymbol{x}_0 \sim p_{\text{data}}, \ \boldsymbol{x}_1 \sim \pi, \tag{3}$$

where $\boldsymbol{w}_t$ is a Wiener process, $g(t) \geq 0$ is a noise schedule, and $\pi$ is a tractable stationary prior (commonly Gaussian). The reverse generative process can be formulated either as a reverse-time SDE or, equivalently, as the *probability-flow ODE* (PF-ODE):

$$\frac{d\boldsymbol{x}_t}{dt} = \mathbf{F}(\boldsymbol{x}_t, t), \qquad \mathbf{F}(\boldsymbol{x}_t, t) = -\dot{\sigma}(t)\, \sigma(t)\, \nabla_{\boldsymbol{x}_t} \log p_t(\boldsymbol{x}_t), \tag{4}$$

where $\mathbf{F}(\boldsymbol{x}_t, t)$ is the deterministic drift vector field whose trajectories have the same marginals as the forward SDE, $\sigma(t)$ parameterizes the forward variance, and $\nabla_{\boldsymbol{x}_t} \log p_t(\boldsymbol{x}_t)$ denotes the score function. Intuitively, $\mathbf{F}$ "denoises" $\boldsymbol{x}_t$ toward $\boldsymbol{x}_0$ as $t$ decreases from 1 to 0. This *deterministic* PF-ODE offers a *single, unique trajectory* that guides samples from noise to data. The PF-ODE thus ties all noise levels together, and one can enforce consistency along the path by matching predictions for all points on that path. This notion of single-path consistency has been found promising to developing powerful models for few-step generation in continuous domain (Song et al., 2023; Song & Dhariwal, 2024), although they remain practically challenging to train (Geng et al., 2025).

**The need for a new consistency formulation in discrete space.** In discrete diffusion, different corruption levels *do not* lie on a unique trajectory. There is no equivalent of a PF-ODE, and hence no canonical map $\boldsymbol{x}_t \mapsto \boldsymbol{x}_s$. This absence has been the primary obstacle to developing a principled consistency framework for discrete data. Our work introduces a conceptual shift: instead of searching for a non-existent deterministic path, we leverage the rich web of *stochastic* paths. Our key observation is that the discrete diffusion framework (Austin et al., 2021a) already provides an analytic family of such paths connecting any two noise levels, which is a powerful yet overlooked property of these diffusion processes. CDLM replaces the missing PF-ODE with these exact posterior bridges and enforces *multi-path discrete consistency*: predictions must agree across many valid routes, making short routes and long routes equivalent in expectation.

## 3 METHOD

We present Consistent Diffusion Language Models (CDLM), a discrete generative framework built on the principle of *Multi-Path Discrete Consistency*. Prior consistency methods in continuous domains rely on the PF-ODE to define a unique, deterministic trajectory connecting noise to data. In discrete space, however, no such ODE exists. Instead of attempting to discretize a non-existent trajectory, we embrace the stochastic nature of discrete diffusion. We observe that the discrete framework defines a rich family of stochastic paths connecting any two noise levels via exact posterior bridges.

Our core insight is to generalize consistency to this stochastic regime: we train a time-conditional predictor to be *path-independent in expectation*. This means that a direct prediction from a highly corrupted state $\boldsymbol{x}_t$ must agree (in expectation) with the prediction made after taking an intermediate "hop" to $\boldsymbol{x}_s$ via any valid stochastic bridge. By enforcing this consistency, CDLM learns to decompose the difficult mapping $\boldsymbol{x}_t \rightarrow \boldsymbol{x}_0$ into arbitrary sub-problems, enabling high-quality generation in few steps as an emergent property of training.

### 3.1 LEARNING A PATH-INDEPENDENT DENOISER

To learn a consistent denoiser, we leverage the analytic reversibility of the discrete process between *arbitrary* timesteps.

**Lemma 1** (General Posterior Bridge). *For any $0 \leq s < t$, the analytic posterior bridge for a single token position is given by:*

$$q(x_s^i \mid x_t^i, x_0^i) = \text{Cat}\left(x_s^i; \frac{(x_0^i \boldsymbol{Q}_{1:s}) \odot (\boldsymbol{Q}_{s+1:t}^\top x_t^i)}{\langle x_0^i \boldsymbol{Q}_{1:t}, \ x_t^i \rangle}\right). \tag{5}$$

*The sequence-level bridge is the product over all positions: $q(\boldsymbol{x}_s|\boldsymbol{x}_t, \boldsymbol{x}_0) = \prod_i q(x_s^i|x_t^i, x_0^i)$. Furthermore, these bridges compose transitively, obeying a semigroup property: for any $u < s < t$, traversing the bridge from $t \to s$ and then $s \to u$ is equivalent to traversing the direct bridge from $t \to u$.*

Using the bridge operator, we can define what it means for a denoising function to be consistent across different paths. We seek to learn a function $f_\theta(\boldsymbol{x}_t, t)$ that predicts the clean data $\boldsymbol{x}_0$ from any noisy input.

**Definition 1** (Multi-path Consistency Operator). *Let $g : \mathcal{X} \times [0, 1] \to \Delta^{|\mathcal{V}|}$ be a time-conditional predictor. The multi-path consistency operator, $C_{s \leftarrow t}$, transforms this function as follows:*

$$\big[C_{s \leftarrow t}g\big](\boldsymbol{x}_t, t; \boldsymbol{x}_0) := \mathbb{E}_{\boldsymbol{x}_s \sim q(\boldsymbol{x}_s|\boldsymbol{x}_t, \boldsymbol{x}_0)}\big[g(\boldsymbol{x}_s, s)\big]. \tag{6}$$

*This operator returns the expected prediction of $g$ at time $s$, after transitioning from time $t$ via the posterior bridge.*

The ideal denoising function would be a fixed point of this operator for all possible timesteps.

**Definition 2** (Global Multi-path Consistency). *A function $f^\star$ is globally multi-path-consistent if for all $0 \le s < t \le 1$, it is a fixed point of the consistency operator in expectation over the data distribution:*

$$f^\star(\boldsymbol{x}_t, t) = \mathbb{E}_{\boldsymbol{x}_0 \sim p(\boldsymbol{x}_0|\boldsymbol{x}_t)}\big[[C_{s \leftarrow t}f^\star](\boldsymbol{x}_t, t; \boldsymbol{x}_0)\big], \tag{7}$$

**Lemma 2** (Optimal Predictor). *Let $f^*(\boldsymbol{x}_t, t) := p(\boldsymbol{x}_0|\boldsymbol{x}_t)$ be the posterior marginal distribution over clean data. Then $f^*$ is a fixed point of the global multi-path consistency condition. Furthermore, under any strictly proper scoring rule $\mathbb{D}$ (e.g., KL divergence, JSD), $f^*$ is the unique minimizer of the expected consistency loss.*

This condition formalizes path-invariance: predicting from $\boldsymbol{x}_t$ directly is equivalent to first transitioning to *any* intermediate state $\boldsymbol{x}_s$ and predicting from there.

**Training.** We train a model $f_\theta$ to satisfy the global consistency property by minimizing the discrepancy between the two sides of Eq. 7 over a random selection of timesteps and data. For a chosen *step size $\delta = t - s$*, the CDLM objective is:

$$\mathcal{L}_{\text{CDLM}}(\theta) = \mathbb{E}_{t,\delta,\boldsymbol{x}_0,\boldsymbol{x}_t,\boldsymbol{x}_s}\left[\sum_{i \in \mathcal{M}(\boldsymbol{x}_t)} w(t, \delta) \cdot \mathbb{D}\left(f_{\theta,i}(\boldsymbol{x}_t, t) \,\Big\|\, f_{\tilde{\theta},i}(\boldsymbol{x}_s, s)\right)\right], \tag{8}$$

where the loss sums over masked positions $\mathcal{M}(\boldsymbol{x}_t)$. Here, $f_{\tilde{\theta}}$ denotes a target network whose parameters $\tilde{\theta}$ are a variant of $\theta$ (e.g., a slow-moving exponential average) to stabilize training. The term $\mathbb{D}$ is a divergence measure between the two output distributions and $w(t, \delta)$ is a positive weighting function. Enforcing local consistency across edges also implies global path-independence, which we formalize in Appendix.

**Sampling.** A trained CDLM is a time-conditional denoiser, analogous to a standard MDLM, which allows it to leverage existing sampling. We use ancestral sampling, where given a sequence $\boldsymbol{x}_t$, we first predict its clean version $\hat{\boldsymbol{x}}_0 = f_\theta(\boldsymbol{x}_t, t)$ and then use the posterior bridge $q(\boldsymbol{x}_s \mid \boldsymbol{x}_t, \hat{\boldsymbol{x}}_0)$ to sample the next state $\boldsymbol{x}_s$ (Austin et al., 2021a; Sahoo et al., 2024). Note that CDLM's novelty lies not in devising new samplers, but in training a model that remains robust under any schedule of steps, although compatibility with existing samplers helps with fair comparison and adoption.

## 3.2 Design Insights for Stable and Scalable Training

While the default CDLM objective in Eq. 8 suffices for simple settings, such as the 2D moons data in Fig. 1, the multipath consistency objective is self-referential, creating an optimization landscape where it takes very long to converge or even convergence degenerate solutions. In particular, naive optimization could lead to *mode collapse*, where the outputs become overly repetitive and deterministic to trivially satisfy consistency, or *uniform drift*, where predictions degrade towards uninformative distributions that are easy to 'match'. We introduce three principled design choices that stabilize training and scale CDLM effectively.

**Step size as a multi-task curriculum.** In CDLM, the step size $\delta = t - s$ determines how far we "jump" along a denoising route. Through linearity of expectation, we can view CDLM training as *multi–task learning over step sizes*: each $\delta$ specifies a distinct path–equivalence constraint. This perspective makes two design questions explicit: (i) *which* step sizes should be practiced (the step size scheduler $p(\delta)$), and (ii) *how* to weight them (the weighting scheduler $w(\delta)$). We sample $\delta$ within a practical range (e.g., $1/8$–$3/8$), which directly targets the few-step regime where efficiency gains matter most. Moreover, we select $w(\delta) = \frac{1}{\delta}$ to help with *path length normalization*, to make each unit of "time" on the corruption axis contribute equally, regardless of whether it is traversed in many short hops or a few long jumps. In practice this choice prevents the training signal from being dominated by easy, local constraints while still supplying dense supervision where it is most stable.

**A diffusion anchor via max-step scheduler.** The self-referential nature of the CDLM loss can be stabilized by grounding it with the true data distribution. We mix in a small fraction of "max-step" tasks where $\delta = t$ (so $s = 0$). The corresponding loss

$$\mathcal{L}_{\text{final}}(\theta) = (1 - \kappa_{\text{ms}}) \, \mathcal{L}_{\text{CDLM}}(\theta) + \kappa_{\text{ms}} \, \mathbb{E}_{t, \boldsymbol{x}_0, \boldsymbol{x}_t} \Big[ \frac{1}{t} \, \mathbb{D}(f_\theta(\boldsymbol{x}_t, t) \, \| \, \boldsymbol{x}_0) \Big] \tag{9}$$

recovers the standard diffusion objective as a regularizer. In practice, a small $\kappa_{\text{ms}} \in [0.1, 0.4]$ suffices to ground learning and discourage low–entropy "shortcut" solutions. Moreover, we find this regularization is most critical in the early stages of training and its weight can be annealed over time.

**Optimization Asymmetry and Choice of Divergence.** To prevent the model from collapsing by perfectly matching its own (potentially flawed) predictions, we introduce an optimization asymmetry. This is implemented using a stop-gradient on the target network, whose parameters are a slow-moving exponential average (EMA) of the online model (Grill et al., 2020). Furthermore, to balance the mode-seeking and mode-covering tendencies of forward and reverse KL-divergence, which can exacerbate collapse and drift respectively, we use the symmetric and bounded Jensen-Shannon Divergence, which provides more stable gradient signal when training from scratch.

## 3.3 A Unifying View of Discrete Generative Modeling

The multi-path consistency principle not only enables efficient generation but also provides a general lens through which to understand and connect a range of modern generative models. We now show that the canonical objectives for masked diffusion, consistency models, and other acceleration techniques emerge as specific instantiations of the CDLM framework.

**Masked Diffusion Models (Sahoo et al., 2024) as the max-step specialization.** CDLM reduces to standard MDLM training by exclusively using the maximum possible step size, $\delta = t$, which sets the prior step to $s = 0$, and collapses the target in CDLM objective to the boundary $f(\boldsymbol{x}_0, 0) = \boldsymbol{x}_0$. With the diffusion weight $w(t) = -\alpha_t'/(1 - \alpha_t) = 1/t = 1/\delta$ (for a linear schedule) and KL-divergence (or equivalently cross-entropy) as distance, Eq. 8 reduces to the standard masked-diffusion NELBO.

**Consistency Models (Song et al., 2023) as the small-step limit with single-path consistency.** In the small step-size limit, $\delta \to 0$, the objective in Eq. 8 concentrates on enforcing local self-consistency across adjacent steps. While continuous Consistency Models rely on a one-step ODE solver to couple adjacent points, CDLM uses the exact posterior bridge, providing a rigorous and native foundation for local consistency training in discrete space.

**Progressive Distillation (Salimans & Ho, 2022) and Shortcut Models (Frans et al., 2025) via the bridge semigroup.** The bootstrap principle underlying both Progressive Distillation and Shortcut Models, that one large step should equal two smaller steps, emerges directly from the semigroup property of the posterior bridge (Lemma 1). Let $T_\delta f$ be the consistency operator. The fixed-point condition $f = T_\delta f$ and the composition rule $T_{2\delta} = T_\delta \circ T_\delta$ together imply that consistency over a step size $\delta$ begets consistency over $2\delta$. Training with a geometric curriculum on $\delta$ thus naturally implements the logarithmic stage progression of PD without an external teacher. Conditioning $f_\theta$ on $\delta$ further recovers the core mechanism of Shortcut Models.

**Discrete distillation as approximate bridge implementations.** CDLM is a *single-stage* training principle that keeps supervision in discrete space via the *analytic* bridge $q(\boldsymbol{x}_s \mid \boldsymbol{x}_t, \boldsymbol{x}_0)$. Recent *two-stage distillation* methods share the goal of few-step generation but realize the bridge *approximately*. Self-Distillation Through Time (SDTT) (Deschenaux & Gulcehre, 2025) constructs adjacent targets by teacher rollouts: starting from $\boldsymbol{x}_t$, a learned teacher applies a few small steps to produce an approximate neighbor $\tilde{\boldsymbol{x}}_{t-\delta}$, effectively using $p_{\text{teacher}}(\boldsymbol{x}_{t-\delta} \mid \boldsymbol{x}_t)$ in place of the analytic $q(\boldsymbol{x}_{t-\delta} \mid \boldsymbol{x}_t, \boldsymbol{x}_0)$, and repeats this in progressive stages. Duo with Discrete Consistency Distillation (DCD) (Sahoo et al., 2025) exploits "diffusion duality" by sharing Gaussian noise across two times and mapping continuous states to the discrete domain via `argmax`. Concretely, if $\epsilon$ is shared, $\boldsymbol{x}_s = \arg\max\big((1-s)\boldsymbol{x}_0 + s\,\epsilon\big)$ and $\boldsymbol{x}_t = \arg\max\big((1-t)\boldsymbol{x}_0 + t\,\epsilon\big)$ form a deterministic adjacent pair; doubling the step per round yields a geometric schedule. This realizes an *algorithmic* bridge (PF-ODE path $\rightarrow$ `argmax` projection) rather than the exact posterior.

## 4 EXPERIMENTS

### 4.1 RELATED BASELINES FOR TEXT GENERATION

We compare our models with MDLM (Sahoo et al. (2024)), SDTT (Deschenaux & Gulcehre (2025)), and DUO (including DUO-DCD)(Sahoo et al. (2025)), which are currently the best models at their scale in terms of generation quality. We will briefly introduce the models we are comparing against here, and leave an additional overview of other less related works in the Appendix section A.1. MDLM is a text based diffusion models with masked distribution as prior and is trained with the NELBO loss. For uniform prior models, DUO improves upon the original Uniform Diffusion Language Models (UDLM, Schiff et al. (2024)) by leveraging a connection to continuous Gaussian distribution through an argmax operation. DUO further distilled their base model with by applying the argmax operator over continuous consistency distillation, namely DUO-DCD and they found using a greedy sampler further improves the sampling metrics. Likewise, SDTT also performs distillation based on MDLM, yet they formulate the distillation process as self-distillation with multistep sampling as the target.

Conceptually, there appears a distinct classification of these models, with one being recognized as the *base model* which is trained with the same objective, while the other belongs to *distilled model* which relies on the base model as teacher and requires single or multiple steps of teacher roll-outs for better generation quality across different sampling steps. Same as MDLM and DUO, our CDLM is trained only with consistency loss from scratch and thus can serve as a base model, while SDTT and DUO-DCD used a completely different distillation-only loss which places themselves into the second category.

### 4.2 EXPERIMENTAL SETUP

We present two models trained with masked source distribution with 110M parameters, namely MCDLM (Masked CDLM) and MCDLM-PPLOptimized. Both models are trained within a single stage for 150K steps using Algorithm 2 with multischeduler objective from Equation 9, and MCDLM-PPLOptimized is a CDLM variant that gives much better generative perplexity with a slightly sacrificed entropy. We compare our models with both categories for unconditional 4.3.1 and conditional 4.3.2 generations.

For MCDLM we set our step size schedulers $\Delta_T$ to be a random scheduler in $[\frac{1}{8}, \frac{5}{8}]$, with $\kappa_{ms}$ for Max-Step Scheduler set to $0.4$. For MCDLM-PPLOptimized, we use the same setting as CDLM for the first 100K steps, then gradually shrinking the max range of $\Delta_T$ to $\frac{3}{8}$, as well as shrinking $\kappa_{ms}$ to 0.2. For stabilized training (Sahoo et al. (2025); Schiff et al. (2024); Song et al. (2023)), we use Exponential Moving Average ($\lambda = 0.999$) for $\bar{\theta}$ during CDLM training, and changed to a hard update for every 10k steps starting starting at 100k steps for MCDLM-PPLOptimized.

To show the generalizability of our algorithm, we also present experiment trained with Uniform Distribution as prior, namely UCDLM and we keep the same data setup. For training, we use a linear increasing step size scheduler from 0.125 to 0.375. We keep the rest of the config same as MCDLM.

Consistent with our models, we train all of the compared models with 110M parameters for 150K steps and a batch size of 2048 using OpenWebTextGokaslan & Cohen (2019). MDLM was trained

| Model | Pretrain Steps | Distill Steps | Sampling steps with FP64 Sampling | | | | | | | | |
|---|---|---|---|---|---|---|---|---|---|---|---|
| | | | 4 | 8 | 16 | 32 | 64 | 128 | 256 | 512 | 1024 |
| *Comparison with Base Models (Trained from Scratch)* | | | | | | | | | | | |
| AR | 75K | 0 | N/A | N/A | N/A | N/A | N/A | N/A | N/A | N/A | 40.2 (5.6) |
| MDLM | 150k | 0 | 1654.5 (5.8) | 682.7 (5.9) | 297.1 (5.6) | 186.9 (5.8) | 124.4 (5.6) | 129.2 (5.8) | 100.5 (5.7) | 114.0 (5.6) | 97.7 (5.6) |
| Ours: MCDLM | 150k | 0 | 649.4 (5.5) | 246.9 (5.6) | 125.6 (5.4) | 86.5 (5.6) | 67.7 (5.6) | 66.0 (5.5) | 55.4 (5.5) | 58.4 (5.4) | 53.4 (5.5) |
| Ours: MCDLM–PPLOptimized | 150k | 0 | 331.2 (5.0) | 132.1 (5.2) | 71.6 (5.3) | 48.7 (5.3) | 40.1 (5.3) | 38.1 (5.2) | 32.5 (5.3) | 33.5 (5.2) | 33.8 (5.4) |
| *Comparison with Distilled Models* | | | | | | | | | | | |
| MDLM - SDTT | 100k | 50k | 369.6 (5.3) | 134.0 (5.3) | 76.0 (5.4) | 51.4 (5.6) | 40.1 (5.3) | 36.2 (5.4) | 32.5 (5.3) | 33.8 (5.1) | 31.2 (5.4) |
| Ours: MCDLM + SDTT | 100k | 50k | **242.1 (5.2)** | **105.0 (5.3)** | **57.5 (5.1)** | **47.0 (5.5)** | **35.3 (5.3)** | **30.3 (5.2)** | **25.8 (5.3)** | **28.1 (4.9)\*** | **27.1 (5.2)** |

Table 1: Generative perplexity (with entropy in parentheses) across different models with *Masked Distribution* as prior which we call MCDLM, training setups, and FP64 sampling steps. We use ancestral sampler for all models. Results with best PPLs are **bolded** and second best are underlined. * denotes the entropy is lower than 5 which we found empirically yield repetitive characters. Consistent with MDLM, our AR baseline is trained with half of the steps to ensure the number of total seen tokens are the same during training.

| Model | Pretrain Steps | Distill Steps | Sampling steps with FP64 Sampling | | | | | | | | |
|---|---|---|---|---|---|---|---|---|---|---|---|
| | | | 4 | 8 | 16 | 32 | 64 | 128 | 256 | 512 | 1024 |
| *Comparison with Base Models (Trained from Scratch)* | | | | | | | | | | | |
| AR | 75K | 0 | N/A | N/A | N/A | N/A | N/A | N/A | N/A | N/A | 40.2 (5.6) |
| UDLM | 150k | 0 | 516.6 (5.5) | 185.9 (5.7) | 122.4 (5.6) | 93.9 (5.7) | 87.6 (5.6) | 90.5 (5.7) | 78.2 (5.7) | 83.1 (5.4) | 84.0 (5.4) |
| DUO | 150k | 0 | 514.4 (5.6) | 177.3 (5.6) | 123.2 (5.4) | 97.7 (5.4) | 85.1 (5.4) | 83.4 (5.5) | 89.4 (5.5) | 91.2 (5.5) | 85.4 (5.6) |
| Ours: UCDLM | 150k | 0 | 377.3 (5.2) | 156.9 (5.7) | 104.7 (5.5) | 85.5 (5.5) | 81.0 (5.5) | 77.6 (5.6) | 74.6 (5.5) | 71.3 (5.4) | 71.7 (5.3) |
| Ours: UCDLM (greedy) | 150k | 0 | 110.4 (4.8)\* | 89.6 (5.6) | 74.4 (5.5) | 65.7 (5.5) | 64.7 (5.5) | 62.1 (5.6) | 58.9 (5.4) | 60.3 (5.6) | 57.0 (5.2) |
| *Comparison with Distilled Models* | | | | | | | | | | | |
| DUO + DCD | 100k | 50k | 408.3 (5.6) | 166.9 (5.6) | 118.2 (5.4) | 91.8 (5.5) | 80.2 (5.5) | 79.4 (5.5) | 77.9 (5.6) | 85.8 (5.6) | 75.6 (5.5) |
| DUO + DCD (greedy) | 100k | 50k | 118.4 (3.9)\* | 109.2 (5.1) | 79.8 (5.1) | 70.5 (5.2) | 65.5 (5.4) | 64.3 (5.3) | 62.6 (5.2) | 67.3 (5.3) | 58.5 (5.1) |

Table 2: Generative perplexity (with entropy in parentheses) across different models with *Uniform Distribution* as prior which we call UCDLM, and FP64 sampling steps. We use ancestral sampler for all models except DUO + DCD (greedy), which uses greedy sampler described as in . Results with best PPLs are **bolded**. * denotes the entropy is lower than 5 which we found empirically yield repetitive characters. Consistent with MDLM, our AR baseline is trained with half of the steps to ensure the number of total seen tokens are the same during training.

with NELBO objective for 150K steps, and SDTT undergoes a pretraining stage of 100K steps using MDLM's objective before shifting to distillation with 2 teacher updates per step for 50K steps. For DUO, similar to Sahoo et al. (2025) we always use half of the steps for curriculum learning and half of the steps for continual finetuning. DUO + DCD leverages the DUO trained with 100K steps and then uses an additional 50K steps for distillation with updating teacher and doubling the delta for every 10K rounds.

### 4.3 RESULTS AND ANALYSIS

#### 4.3.1 UNCONDITIONAL GENERATION

We present results for unconditional generation with 1024 tokens in Figure 2 and Table 1 for 64-bit sampling and Table 4 for 32-bit sampling. Each model generates 32 samples for PPL evaluation under `gpt2-large`. Our CDLM model outperforms MDLM for all of the sampling steps, regardless of the sampling precision. Compared to DUO, CDLM produces much lower PPLs through step 16 to 1024 under both 32 and 64-bit sampling, while maintaining a similar entropy as DUO under 64-bit sampler. Moreover, comparing to distilled models which are generally better in PPL with lower entropy as a result of distillation, CDLM-PPLOptimized is trained without the distillation stage but still outperforms multistage models like SDTT and DUO-DCD with both ancestral and greedy samplers for most of the sampling steps, while keeping the same entropy level. Note that although DUO-DCD with greedy samplers returns the lower PPL for at low sampling steps, it actually also suffers from significantly lower entropy which indicates mode collapsing that produces repetitive characters which could also hack the metrics. We also distilled CDLM using SDTT objective, which gives a model outperforming the original SDTT throughout step 4 to 1024 under both 32 and 64-bit sampling. Overall, CDLM-PPLOptimized achieves the best balance between generative PPL and entropy. Speedwise, CDLM–PPLOptimized is able to achieve a 64x-128x speedup compred to MDLM. Comparing with the AR model, CDLM–PPLOptimized was able to achieve similar

| Model | OpenWebText | | | Lambada | | | Wikitext103 | | | PTB | | |
|---|---|---|---|---|---|---|---|---|---|---|---|---|
| | 8 Step | 64 Step | 512 Step | 8 Step | 64 Step | 512 Step | 8 Step | 64 Step | 512 Step | 8 Step | 64 Step | 512 Step |
| MDLM | 45.9 / 60.9 | 40.4 / 61.1 | 39.7 / 61.1 | 72.3 / 57.6 | 62.9 / 57.7 | 61.7 / 57.9 | 44.8 / 61.9 | 39.6 / 62.2 | 39.0 / 62.2 | 248.2 / 50.5 | 220.0 / 50.7 | 215.8 / 50.6 |
| SDTT | 34.0 / 62.4 | 30.8 / 62.5 | 30.3 / 62.5 | 51.9 / 59.3 | 46.4 / 59.3 | 45.7 / 59.5 | 33.6 / 63.5 | 30.5 / 63.7 | 30.1 / 63.7 | 141.4 / 52.3 | 125.4 / 52.4 | 124.0 / 52.3 |
| Ours: MCDLM–PPLOptimized | **31.6 / 62.8** | **28.6 / 62.9** | **27.9 / 63.1** | **43.4 / 60.1** | **38.8 / 60.1** | **38.5 / 60.3** | **30.0 / 64.2** | **27.3 / 64.4** | **26.9 / 64.5** | **122.1 / 53.2** | **107.5 / 53.3** | **106.8 / 53.3** |
| DUO-DCD (Greedy) | 40.2 / 26.7 | 32.7 / 26.6 | 32.0 / 26.7 | 27.9 / 31.2 | 31.2 / 59.3 | 21.6 / 31.0 | 44.9 / 32.3 | 35.9 / 32.5 | 34.1 / 32.2 | 62.1 / 19.5 | 45.0 / 19.3 | 41.2 / 19.2 |

Table 3: Conditional Generation results for across different datasets. Perplexity $\downarrow$ / BLEU2 $\uparrow$ results with FP64 sampling using the ancestral sampler are reported. We choose the best performing models from unconditional generation for our comparison. Results for DUO-DCD with greedy sampler are grayed out as it produces nearly random sentences that do not preserve the input conditions, which reults in very low BLEU scores.

performance with between step 32 and 64, thus offering a 16-32x speedup in terms of Number of Function Evaluations (NFE).

We observe similar success of our UCDLM model, which was to apply 1 to uniform prior diffusion language models. As shown in 2, our UCDLM model outperforms vanilla UDLM across all sampling steps. Moreover, comparing to the state-of-the-art DUO Sahoo et al. (2025) model and its distillation variant, UCDLM also yields better perplexity regardless of the sampler choice (ancestral or greedy).

### 4.3.2 CONDITIONAL GENERATION

We also evaluate our model on conditional generation across four popular datasets including three out-of-distribution sets: OWT (Gokaslan & Cohen (2019)), Lambada (Paperno et al. (2016)), Wikitext2 (Merity et al. (2016)), and PTB (Marcus et al. (1993)). We randomly sample 32 sentences with 1024 tokens from each dataset, purturring 50% of the tokens with the model's prior distribution, and serve them as conditions given to the model to recover the original sentences. We use the original unperturbed sentences as reference, with PPL evaluating the fluency of the final generated sentence as well as BLEU assessing if the model is able to conditionally generate sentences plausible as the reference sentences. In additional to token-space similarity, we also use MAUVE (Pillutla et al. (2021)) to evaluate the embedding space distribution matching, although this metric is over saturated for our task so we put it into Table 5 in Appendix. Table 3 outlines the comparison of our CDLM versus SDTT and DUO. Given its uniform distribution formulation which allows tokens to transit into any other tokens during sampling, DUO is not good as preserving the conditions and often yield very low BLEU score and generate sentences that completely differs from the given condition, so therefore we grey it out. Again, CDLM-PPLOptimized consistenctly outperforms SDTT in terms of generation perplexity and BLEU score, demonstrating its advantage in generating plausible, consistent and fluent sentences under given conditions.

### 4.4 ABLATIONS AND INSIGHTS

We conduct ablation studies to show the effectiveness of

**Choice of distance metric** We illustrate the reason behind the use of JS-divergence as the distance metric for CDLM training. The mode-seeking forward KL objective results in a sharp drop in entropy after certain steps, and the samples show repetitive characters despite have better perplexities. The mode-covering backward KL objective, on the other hand, results in very high entropy as well as very high perplexities. DUO-DCD and SDTT do not suffer from the KL issues mostly because they already have a very strong teacher model that avoids model collapsing and uniform drift, while our model suffers as we are training from scratch. For more details please see Table 8 in Appendix.

**Max-step scheduler and diffusion regularizer** We empirically found that incorporating max-step scheduler helps with balancing the generation quality and diversity. We observe that without using the max-step scheduler, our model is quickly optimized towards mode-collapsing, producing repetitive words with very low entropy as well as low but biased perplexity. With increasing weights for diffusion regularizer, we observe that both PPL and Entropy increases across all steps, indicating diffusion regularizer as balancer between diversity and quality. More details in Table 7 in Appendix.

**Choice of step size scheduler** Other than the max-step scheduler serving as diffusion regularizer, we also use a separate scheduler for $t$ and $\delta$ for the CDLM training. Note that training without any

sampler other than diffusion regularizer reduces our model to MDLM. We experimented with four schedulers: random, linear increasing, and linear decreasing. Model trained with linear increasing scheduler got exposed to small $\delta$ at the beginning and bigger ones towards the end, making the later checkpoints better at generation with more steps. Likewise, models trained with linear decreasing scheduler tends to do better generation with few-steps. For more details, see Table 6 in Appendix.

## 5 DISCUSSION AND CONCLUSION

We introduced the Consistent Diffusion Language Model (CDLM), a new framework for discrete generative modeling built on the principle of multi-path consistency. By supervising with exact posterior bridges, CDLM trains a path-independent denoiser that achieves few-step efficiency as a training-time property rather than a post-hoc acceleration. The result is a single-stage model that advances the state of scalable, high-fidelity text generation.

**Synergies with Complementary Advances.** While CDLM illustrates strong algorithmic efficiency, measured in sampling steps, diffusion LMs still incur high wall-clock latency *per step* than optimized autoregressive decoders. This remains a key deployment challenge. Crucially, CDLM is architecture- and sampler-agnostic, and can directly benefit from advances such as KV-caching, optimized kernels, and faster sampling schemes. Recent works demonstrating over $10\times$ speedups in diffusion inference (Ma et al., 2025a; Wu et al., 2025; Liu et al., 2025a) suggests that CDLM's efficiency gains can be amplified through such engineering advances.

**The Design Space of Multi-Path Consistency.** CDLM should be understood not as a fixed algorithm, but as a flexible framework with a rich set of design choices. Our implementation explores one principled configuration, yet many alternatives remain, including adaptive step schedules, alternative weighting schemes, or different divergence metrics. The rapid evolution of continuous consistency models (Song & Dhariwal, 2024; Geng et al., 2025) through similar refinements suggests that CDLM is a promising starting point with significant potential for further gains.

**A Foundation for Future Models.** An important implication of CDLM is its role as a stronger base model for the next generation of discrete generative methods. Many leading acceleration techniques, such as distillation, build on pre-trained base models. We show that CDLM outperforms MDLM as such a foundation, and it can serve as a promising replacement for large-scale pretraining or post-training mechanisms with downstream benefits (Nie et al., 2025b). More broadly, the formulation is not tied to language alone. Any domain involving discrete structures, such as biological sequence or program synthesis, can benefit from this framework.

In reframing discrete diffusion as the training of a path-independent denoiser, CDLM bridges the gap between the acceleration playbooks of continuous diffusion and the realities of discrete data. We hope this work not only advances the frontier of few-step discrete generation, but also lays the foundation for models that are fast, principled, and broadly applicable.

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

# A APPENDIX

## A.1 RELATED WORK

**Discrete Diffusion/Flow Models** Austin et al. (2021b); Lou et al. (2024) introduced diffusion models for discrete data, followed by MDLM (Sahoo et al. (2024)) showing initial success on text modeling. Our work is based on MDLM, a masked diffusion language model trained on NELBO objective that is simplified as a time-weighted cross-entropy loss. Discrete Flow Matching (Gat et al. (2024)) formulates text generation under flow-matching perspective optimizing for the learned marginal velocity field, yielding a similar training objective as MDLM under the masked prior. Beside masked prior distribution, UDLM (Schiff et al. (2024)) and DUO (Sahoo et al. (2025)) introduced and improved the uniform prior diffusion models, unlocking better generation quality by leveraging the unique advantage of uniform transition kernel, which includes guided training and sampling, as well as discretization of continuous Gaussian distribution respectively. CDLM shares commonalities with these models in that they all require single stage training with a unified objective, yet CDLM delivers better generation quality through our formulation of discrete consistency training.

**Improving few step generation for diffusion models.** Currently there are two main fields of accelerating diffusion language model, where one focuses on training-free acceleration including the use of KV Cache (Ma et al. (2025b;a); Liu et al. (2025b)), as well as alternative sampling and decoding strategies (Chen et al. (2025); Huang et al. (2025); Ben-Hamu et al. (2025); Gwak et al. (2025)). The other field focuses on training based approach that relies mainly on distillation from a pretrained model. For instance, DUO w. DCD (Sahoo et al. (2025)) operates on a pretrained DUO model, and applies consistency loss with $x_s$ and $x_t$ sampled from discretized Gaussian Path. SDTT (Deschenaux & Gulcehre (2025)) operates on a pretrained MDLM model, requiring multiple steps of teacher rollouts as the training target. We have also shown in Section 3.3 that both DUO and SDTT falls within a special case of CDLM which empirically shows better generation quality.

**Consistency model family** Consistency models (Song et al. (2023)) were first proposed for image generation in continuous domain, with later works (Song & Dhariwal (2024); Geng et al. (2025)) improving the training in terms of simplicity as well as performance. DUO successfully adopts consistency distillation to become better at few-step generation through connecting uniform discrete diffusion with continuous gaussian distribution. Additionally, CDLM extends this concept to allow training from scratch with non-uniform diffusion models.

## A.2 ALGORITHM

---

**Algorithm 1** Consistent Discrete Denoising Diffusion Training (CD3T)

---

**Require:** Dataset $\mathcal{D}$, initial model weights $\theta_0$, weighting function $w(t)$, step size scheduler $\Delta_{1:T}$, EMA scheduler $\lambda$
**Ensure:** Trained model parameters $\theta$
1: **Initialize:** $\theta \leftarrow \theta_0, \bar{\theta} \leftarrow \theta$
2: **for** each $\delta_i \sim \Delta_{1:T}$ **do**
3:     Sample timestep $t \sim p(t)$
4:     Compute $s \leftarrow t - \delta_i$ where $s \sim p(s \mid t, \delta_i)$
5:     Sample data point $\boldsymbol{x}_0 \sim \mathcal{D}$
6:     Sample forward process: $\boldsymbol{x}_t \sim q(\boldsymbol{x}_t \mid \boldsymbol{x}_0) = \mathrm{Cat}(\boldsymbol{x}_t; \boldsymbol{x}_0 \boldsymbol{Q}_{1:t})$
7:     Sample intermediate state: $\boldsymbol{x}_s \sim q(\boldsymbol{x}_s \mid \boldsymbol{x}_t, \boldsymbol{x}_0) = \mathrm{Cat}\left(\boldsymbol{x}_s; \frac{\boldsymbol{x}_t \boldsymbol{Q}_{s+1:t} \odot \boldsymbol{x}_0 \boldsymbol{Q}_s}{\boldsymbol{x}_0 \boldsymbol{Q}_t \boldsymbol{x}_t^\top}\right)$
8:     Compute consistency loss: $\mathcal{L}(\theta, \bar{\theta}) = w(t, \delta_i) \cdot \mathbb{D}\left(f_\theta(\boldsymbol{x}_t, t), f_{\bar{\theta}}(\boldsymbol{x}_s, s)\right)$
9:     Update $\theta$: $\theta \leftarrow \theta - \eta \nabla_\theta \mathcal{L}(\theta, \bar{\theta})$
10:     Update $\bar{\theta}$: $\bar{\theta} \leftarrow \lambda\bar{\theta} + (1 - \lambda)\theta$
11: **end for**
12: **return** $\theta$

---

## A.3 THEORY

**Lemma 1** (General Posterior Bridge). *For any $0 \le s < t$, the analytic posterior bridge is given by:*

$$q(\boldsymbol{x}_s \mid \boldsymbol{x}_t, \boldsymbol{x}_0) \;=\; \mathrm{Cat}\left(\boldsymbol{x}_s; \; \frac{(\boldsymbol{x}_0 \boldsymbol{Q}_{1:s}) \odot (\boldsymbol{Q}_{s+1:t}^\top \boldsymbol{x}_t)}{\langle \boldsymbol{x}_0 \boldsymbol{Q}_{1:t}, \; \boldsymbol{x}_t \rangle}\right). \tag{10}$$

*Furthermore, these bridges compose transitively, obeying a semigroup property: for any $u < s < t$, traversing the bridge from $t \to s$ and then $s \to u$ is equivalent to traversing the direct bridge from $t \to u$.*

*Proof.* We seek to derive the probability vector for the categorical distribution $q(\boldsymbol{x}_s \mid \boldsymbol{x}_t, \boldsymbol{x}_0)$.

**1. Application of Bayes' Rule.** From the definition of conditional probability, we have:

$$q(\boldsymbol{x}_s \mid \boldsymbol{x}_t, \boldsymbol{x}_0) = \frac{q(\boldsymbol{x}_t \mid \boldsymbol{x}_s, \boldsymbol{x}_0) \, q(\boldsymbol{x}_s \mid \boldsymbol{x}_0)}{q(\boldsymbol{x}_t \mid \boldsymbol{x}_0)}$$

---

**Algorithm 2** Masked Consistent Diffusion Language Model

---

**Require:** Dataset $\mathcal{D}$, initial model weights $\theta_0$, step size scheduler $\Delta_{1:T}$, EMA scheduler $\lambda$
**Ensure:** Trained model parameters $\theta$
1: **Initialize:** $\theta \leftarrow \theta_0$
2: **for** each $\Delta_t$ in $\Delta_{1:T}$ **do**
3:     Sample timestep $t \sim \mathcal{U}\{[\Delta_t, 1]\}$
4:     Compute $s \leftarrow t - \Delta_t$
5:     Sample sequence $\boldsymbol{x}_0 = (\boldsymbol{x}_0^1, \ldots, \boldsymbol{x}_0^L) \sim \mathcal{D}$ where $\boldsymbol{x}_0^i \in \mathcal{V}$
6:     Sample corrupted sequence: $\boldsymbol{x}_t \sim q(\boldsymbol{x}_t \mid \boldsymbol{x}_0)$

$$\text{where } q(\boldsymbol{x}_t^i = \boldsymbol{k} \mid \boldsymbol{x}_0^i) = \begin{cases} 1 - t & \text{if } \boldsymbol{k} = \boldsymbol{x}_0^i \\ t & \text{if } \boldsymbol{k} = \texttt{[MASK]} \\ 0 & \text{otherwise} \end{cases}$$

7:     Sample intermediate state: $\boldsymbol{x}_s \sim q(\boldsymbol{x}_s \mid \boldsymbol{x}_t, \boldsymbol{x}_0)$

$$\text{where } q(\boldsymbol{x}_s^i = \boldsymbol{k} \mid \boldsymbol{x}_t^i, \boldsymbol{x}_0^i) = \begin{cases} 1 & \text{if } \boldsymbol{x}_t^i \neq \texttt{[MASK]} \text{ and } \boldsymbol{k} = \boldsymbol{x}_t^i \\ \frac{t-s}{t} & \text{if } \boldsymbol{x}_t^i = \texttt{[MASK]} \text{ and } \boldsymbol{k} = \boldsymbol{x}_0^i \\ \frac{s}{t} & \text{if } \boldsymbol{x}_t^i = \texttt{[MASK]} \text{ and } \boldsymbol{k} = \texttt{[MASK]} \\ 0 & \text{otherwise} \end{cases}$$

8:     Compute consistency loss: $\mathcal{L}(\theta) = \frac{1}{\Delta_t} \cdot \mathbb{D}_{\text{JSD}}\big(f_\theta(\mathbf{x}_t) \,\|\, f_{\tilde{\theta}}(\mathbf{x}_s)\big)$
9:     Update parameters: $\theta \leftarrow \theta - \eta \nabla_\theta \mathcal{L}(\theta)$
10:    Update $\tilde{\theta}$: $\tilde{\theta} \leftarrow \lambda \tilde{\theta} + (1 - \lambda)\theta$
11: **end for**
12: **return** $\theta$

---

**2. Markov Property.** The forward process is a Markov chain, meaning the state at time $t$ depends only on the state at time $s$ (for $s < t$), not on earlier states like $\boldsymbol{x}_0$. Therefore, the likelihood term simplifies:

$$q(\boldsymbol{x}_t \mid \boldsymbol{x}_s, \boldsymbol{x}_0) = q(\boldsymbol{x}_t \mid \boldsymbol{x}_s)$$

This gives us the proportional relationship:

$$q(\boldsymbol{x}_s \mid \boldsymbol{x}_t, \boldsymbol{x}_0) \propto q(\boldsymbol{x}_t \mid \boldsymbol{x}_s) \, q(\boldsymbol{x}_s \mid \boldsymbol{x}_0)$$

**3. Vector Formulation.** We now express the terms on the right-hand side using their categorical probability vectors. Let $\boldsymbol{p}(\cdot)$ denote the probability vector of a distribution.

- The prior probability of $\boldsymbol{x}_s$ is given by the forward marginal: $\boldsymbol{p}(\boldsymbol{x}_s \mid \boldsymbol{x}_0) = \boldsymbol{x}_0 \boldsymbol{Q}_{1:s}$.

- The likelihood of $\boldsymbol{x}_t$ given $\boldsymbol{x}_s$ is determined by the transitions from $s$ to $t$. The probability vector is $\boldsymbol{p}(\boldsymbol{x}_t \mid \boldsymbol{x}_s) = \boldsymbol{x}_s \boldsymbol{Q}_{s+1:t}$.

The expression $q(\boldsymbol{x}_t \mid \boldsymbol{x}_s) \, q(\boldsymbol{x}_s \mid \boldsymbol{x}_0)$ gives the joint probability $q(\boldsymbol{x}_t, \boldsymbol{x}_s \mid \boldsymbol{x}_0)$. To find the probability vector for $q(\boldsymbol{x}_s \mid \boldsymbol{x}_t, \boldsymbol{x}_0)$, we consider the probability of a specific one-hot vector outcome for $\boldsymbol{x}_s$. This is proportional to the probability of that outcome under the prior, multiplied by the probability of observing $\boldsymbol{x}_t$ given that outcome. In vector form, this product corresponds to an element-wise (Hadamard) product of the prior probability vector and the likelihood vector.

The likelihood vector, representing $p(\boldsymbol{x}_t \mid \boldsymbol{x}_s = v_i)$ for all possible states $v_i$, is given by $\boldsymbol{Q}_{s+1:t}^\top \boldsymbol{x}_t$. Thus, the unnormalized probability vector for $\boldsymbol{x}_s$ is:

$$\boldsymbol{p}_{\text{unnormalized}}(\boldsymbol{x}_s \mid \boldsymbol{x}_t, \boldsymbol{x}_0) = (\boldsymbol{x}_0 \boldsymbol{Q}_{1:s}) \odot (\boldsymbol{Q}_{s+1:t}^\top \boldsymbol{x}_t)$$

**4. Normalization.** The normalizing constant is the marginal probability of the evidence, $q(\boldsymbol{x}_t \mid \boldsymbol{x}_0)$. For the specific observed outcome $\boldsymbol{x}_t$, this probability is $\langle \boldsymbol{x}_0 \boldsymbol{Q}_{1:t}, \boldsymbol{x}_t \rangle$. Dividing the unnormalized vector by this scalar gives the final probability vector, completing the proof for the main formula.

**5. Semigroup Property.** The transitive composition follows from the law of total probability and the Markov property:

$$q(\boldsymbol{x}_u \mid \boldsymbol{x}_t, \boldsymbol{x}_0) = \sum_{\boldsymbol{x}_s} q(\boldsymbol{x}_u, \boldsymbol{x}_s \mid \boldsymbol{x}_t, \boldsymbol{x}_0)$$

$$= \sum_{\boldsymbol{x}_s} q(\boldsymbol{x}_u \mid \boldsymbol{x}_s, \boldsymbol{x}_t, \boldsymbol{x}_0)\, q(\boldsymbol{x}_s \mid \boldsymbol{x}_t, \boldsymbol{x}_0)$$

$$= \sum_{\boldsymbol{x}_s} q(\boldsymbol{x}_u \mid \boldsymbol{x}_s, \boldsymbol{x}_0)\, q(\boldsymbol{x}_s \mid \boldsymbol{x}_t, \boldsymbol{x}_0)$$

Substituting the bridge formula (Eq. 10) for each term and simplifying demonstrates that the composition holds, relying on the associativity of the transition matrices ($\boldsymbol{Q}_{u+1:s}\boldsymbol{Q}_{s+1:t} = \boldsymbol{Q}_{u+1:t}$). $\qquad\square$

The following result formalizes the intuition that enforcing local consistency provides a foundation for achieving global path-independence.

**Lemma 2** (From Local to Global Consistency). *Let $\mathbb{D}(\cdot, \cdot)$ be a distance metric that satisfies the triangle inequality (i.e., a norm). For a time grid $1 = \tau_K > \cdots > \tau_0 = 0$, if the expected local consistency error for any one-step transition is uniformly bounded by $\varepsilon$,*

$$\mathbb{E}_{\boldsymbol{x}_0, \boldsymbol{x}_{\tau_k}}\, \mathbb{D}\left( f(\boldsymbol{x}_{\tau_k}, \tau_k),\, \mathbb{E}_{\boldsymbol{x}_{\tau_{k-1}} \sim q(\cdot|\boldsymbol{x}_{\tau_k}, \boldsymbol{x}_0)} f(\boldsymbol{x}_{\tau_{k-1}}, \tau_{k-1}) \right) \le \varepsilon \quad \text{for all } k \in \{1, \ldots, K\},$$

*then the global error between any two points $\tau_m$ and $\tau_K$ on the grid is linearly bounded by:*

$$\mathbb{E}_{\boldsymbol{x}_0, \boldsymbol{x}_{\tau_K}}\, \mathbb{D}\left( f(\boldsymbol{x}_{\tau_K}, \tau_K),\, \mathbb{E}_{\boldsymbol{x}_{\tau_m} \sim q(\cdot|\boldsymbol{x}_{\tau_K}, \boldsymbol{x}_0)} f(\boldsymbol{x}_{\tau_m}, \tau_m) \right) \le (K - m)\varepsilon.$$

*Proof.* The proof proceeds by a recursive application of the triangle inequality, leveraging the convexity of norms and the law of total expectation. For clarity, let $f_k \equiv f(\boldsymbol{x}_{\tau_k}, \tau_k)$ and denote the conditional expectation operator as $E_{j|i}[\cdot] \equiv \mathbb{E}_{\boldsymbol{x}_{\tau_j} \sim q(\cdot|\boldsymbol{x}_{\tau_i}, \boldsymbol{x}_0)}[\cdot]$. The semigroup property of the bridge (Lemma 1) implies that the expectation of the target can be written as a telescoping conditional expectation:

$$\mathbb{E}_{\boldsymbol{x}_{\tau_m} \sim q(\cdot|\boldsymbol{x}_{\tau_K}, \boldsymbol{x}_0)}[f_m] = E_{K-1|K} \circ E_{K-2|K-1} \circ \cdots \circ E_{m|m+1}[f_m].$$

Let $\mathcal{E}(k, m) = \mathbb{E}_{\boldsymbol{x}_0, \boldsymbol{x}_{\tau_k}}\left[ \mathbb{D}\left( f_k, E_{m|k}[f_m] \right) \right]$ be the expected global error from step $k$ to $m$. We wish to bound $\mathcal{E}(K, m)$.

We establish a recursive bound. Consider the error from step $k$ to $m$:

$$\mathbb{D}\left( f_k, E_{m|k}[f_m] \right) = \mathbb{D}\left( f_k, E_{k-1|k}[E_{m|k-1}[f_m]] \right)$$

$$\le \mathbb{D}\left( f_k, E_{k-1|k}[f_{k-1}] \right) + \mathbb{D}\left( E_{k-1|k}[f_{k-1}], E_{k-1|k}[E_{m|k-1}[f_m]] \right) \quad \text{(Triangle Inequality)}$$

$$\le \mathbb{D}\left( f_k, E_{k-1|k}[f_{k-1}] \right) + E_{k-1|k}\left[ \mathbb{D}\left( f_{k-1}, E_{m|k-1}[f_m] \right) \right] \quad \text{(Jensen's Inequality)}$$

The second step uses the fact that a norm $\mathbb{D}$ is a convex function, so for a random variable $X$, $\mathbb{D}(E[X]) \le E[\mathbb{D}(X)]$. Here, we apply it to the outer expectation $E_{k-1|k}$.

Now, we take the expectation $\mathbb{E}_{\boldsymbol{x}_0, \boldsymbol{x}_{\tau_k}}$ over the entire inequality:

$$\mathcal{E}(k, m) \le \mathbb{E}_{\boldsymbol{x}_0, \boldsymbol{x}_{\tau_k}}\left[ \mathbb{D}\left( f_k, E_{k-1|k}[f_{k-1}] \right) \right] + \mathbb{E}_{\boldsymbol{x}_0, \boldsymbol{x}_{\tau_k}}\left[ E_{k-1|k}\left[ \mathbb{D}\left( f_{k-1}, E_{m|k-1}[f_m] \right) \right] \right]$$

$$\le \varepsilon + \mathbb{E}_{\boldsymbol{x}_0, \boldsymbol{x}_{\tau_{k-1}}}\left[ \mathbb{D}\left( f_{k-1}, E_{m|k-1}[f_m] \right) \right]$$

$$= \varepsilon + \mathcal{E}(k - 1, m)$$

We have established the recursive relationship $\mathcal{E}(k, m) \le \varepsilon + \mathcal{E}(k - 1, m)$. By unrolling this relationship from $k = K$ down to $m + 1$:

$$\mathcal{E}(K, m) \le \varepsilon + \mathcal{E}(K - 1, m)$$

$$\le \varepsilon + (\varepsilon + \mathcal{E}(K - 2, m))$$

$$\le \cdots$$

$$\le (K - m)\varepsilon + \mathcal{E}(m, m)$$

Since $\mathcal{E}(m, m) = \mathbb{E}[\mathbb{D}(f_m, E_{m|m}[f_m])] = \mathbb{E}[\mathbb{D}(f_m, f_m)] = 0$, the final bound is:

$$\mathcal{E}(K, m) \le (K - m)\varepsilon.$$

This concludes the proof. $\qquad\square$

## A.4 EXPERIMENTS

| Model | Pretrain | Distill | Sampling steps with FP32 Sampling | | | | | | | | |
|---|---|---|---|---|---|---|---|---|---|---|---|
| | Steps | Steps | 4 | 8 | 16 | 32 | 64 | 128 | 256 | 512 | 1024 |
| *Comparison with Base Models (Trained from Scratch)* | | | | | | | | | | | |
| AR | 75K | 0 | N/A | N/A | N/A | N/A | N/A | N/A | N/A | N/A | 39.9 (5.4) |
| MDLM | 150k | 0 | 1655.2 (5.9) | 651.8 (5.9) | 255.2 (5.8) | 162.3 (5.6) | 92.1 (5.6) | 78.6 (5.4) | 57.5 (5.5) | 51.1 (5.3) | 42.4 (5.4) |
| DUO | 150k | 0 | 532.4 (5.6) | 199.6 (5.6) | 127.3 (5.7) | 96.1 (5.4) | 79.1 (5.5) | 82.4 (5.5) | 78.2 (5.4) | 73.9 (5.5) | 74.8 (5.5) |
| Ours: CDLM | 150k | 0 | 661.1 (5.6) | 220.8 (5.4) | 118.3 (5.6) | 72.6 (5.6) | 56.3 (5.4) | 54.9 (5.4) | 35.2 (5.3) | 29.3 (5.3) | 25.7 (5.2) |
| Ours: CDLM – OptimalPPL | 150k | 0 | 337.1 (5.2) | 117.6 (5.1) | 68.1 (5.2) | 42.4 (5.3) | 35.1 (5.2) | 24.7 (4.9) | 23.6 (5.3) | 21.2 (5.0) | 17.1 (5.2) |
| *Comparison with Distilled Models* | | | | | | | | | | | |
| MDLM + SDTT 150k | 100k | 50k | 351.3 (5.3) | 132.5 (5.5) | 65.7 (5.2) | 44.5 (5.0) | 34.3 (5.3) | 29.9 (5.0) | 24.3 (5.0) | 21.2 (5.0) | 20.8 (4.9)* |
| DUO + DCD | 100k | 50k | 417.0 (5.4) | 172.0 (5.5) | 125.4 (5.6) | 96.2 (5.6) | 81.7 (5.5) | 85.5 (5.5) | 74.1 (5.7) | 72.3 (5.4) | 74.1 (5.3) |
| DUO + DCD (greedy) | 100k | 50k | **127.2 (4.6)*** | **111.0 (4.8)*** | 86.6 (5.0) | 72.6 (5.3) | 62.5 (5.3) | 59.8 (5.2) | 67.2 (5.4) | 61.4 (5.2) | 62.7 (5.2) |
| Ours: CDLM + SDTT | 100k | 50k | 235.9 (5.1) | 94.3 (5.3) | **52.1 (5.2)** | **35.6 (5.0)** | **29.0 (5.2)** | **26.0 (4.9)** | **21.2 (5.0)** | **18.1 (5.1)** | **15.7 (4.6)*** |

Table 4: Perplexity (with entropy in parentheses) across different models, training setups, and FP32 sampling steps. Best results are **bolded**, second-best are underlined. * denotes entropy $< 5$, which empirically led to repetitive characters.

| Model | OpenWebText | | | Lambada | | | Wikitext103 | | | PTB | | |
|---|---|---|---|---|---|---|---|---|---|---|---|---|
| | 8 Step | 64 Step | 512 Step | 8 Step | 64 Step | 512 Step | 8 Step | 64 Step | 512 Step | 8 Step | 64 Step | 512 Step |
| MDLM | 0.90 | 0.94 | 0.96 | 0.99 | 0.99 | 0.98 | 0.94 | 0.96 | 0.97 | 0.11 | 0.12 | 0.10 |
| SDTT | 0.96 | 0.96 | 0.98 | 0.99 | 0.99 | 0.99 | 0.98 | 0.99 | 0.96 | 0.07 | 0.08 | 0.07 |
| Ours: CDLM – OptimalPPL | 0.99 | 0.97 | 0.98 | 0.99 | 0.99 | 0.99 | 0.98 | 0.99 | 0.99 | 0.02 | 0.07 | 0.07 |

Table 5: MAUVE ↑ scores with ancestral sampler across datasets. Given MAUVE is a distribution based metrics and with $50\%$ of unperturbed tokens as the condition, the models perform similarly under this saturated metric.

| Model | 8 | 64 | 1024 |
|---|---|---|---|
| CDLM w. random scheduler | 110.6 / 5.3 | 32.8 / 5.3 | 19.7 / 5.2 |
| CDLM w. staged increasing scheduler | 160.8 / 5.2 | 45.1 / 5.3 | 25.3 / 5.2 |
| CDLM w. linear increasing scheduler | 117.6 / 5.5 | 37.9 / 5.4 | 17.9 / 4.9 |
| CDLM w. linear decreasing scheduler | 112.3 / 5.1 | 32.1 / 5.2 | 20.5 / 5.2 |

Table 6: FP32 Sampling under different scheduling strategies.

| Model | 8 | 64 | 1024 |
|---|---|---|---|
| CDLM w. no max-step scheduler | 16.7 / 3.2 | 7.1 / 3.4 | 5.7 / 2.8 |
| CDLM w. 0.4 for max-step scheduler | 110.6 / 5.3 | 32.8 / 5.3 | 19.7 / 5.2 |
| CDLM w. 1.0 for max-step scheduler | 274.0 / 5.5 | 75.1 / 5.5 | 38.4 / 5.3 |

Table 7: FP32 Sampling under different max-step scheduler/diffusion regularizer weights.

| Model | 8 | 64 | 1024 |
|---|---|---|---|
| CDLM w. JS divergence | 110.6 / 5.3 | 32.8 / 5.3 | 19.7 / 5.2 |
| CDLM w. Forward KL | 8e4 / 6.9 | 8e4 / 6.9 | 7e4 / 6.8 |
| CDLM w. Backward KL | 75.3 / 4.6 | 64.0 / 4.4 | 44.2 / 4.3 |

Table 8: FP32 Sampling under different divergence objectives.

## A.5 QUALITATIVE EXAMPLES

We present qualitative results of generated texts for different models under different sampling steps.

**Step 4**:
MDLM:
```
<|endoftext|>abad in Texas Cordalo.  Gold Day\n \nThere weren
```

very many test there of a citizen in or recognition a Zina which
directly would they need for over 49 little of LRM, American
valhouettes would thinkBut there are other sinister were trying
said 5560 July Saints shipped outed, came out and something
(afterne 73 LA, EEAN) was and and really a louder cagebl hundred
drove Greensboro four pissed, sacked away the same or of the
earlier Sony and large feasible, statement later made it just
didn't come out impoverished means toave.be joiningTexas,
besidesasca curator book is nearly a subscriber and daily
editor within and outside LSUs.  be to making another alum soon
longtime never in the been its preferred basketball and ever
was:bill speaker of Louisianas industry, one of its most poorly
conductedacted jobs in the , toosa state of becoming.Ste blocsed
across same guy on the Tuesday night naive and about Lockntaking
alerted to Ruff.  (Also Second does writing the finddesigne (line
truncated to 1000 characters)

DUO+DCD:

"_\" would slip or top out 90 means how much these parts comes
face .  what's not even more or much far a pretty far way make
the times be manufactd or far tried far ago much much go pretty
well or'and_belso the more side rather other than much in almost
Franks sake.  \" or sversely, look in Archre at least or much
the more of work much noJaAP .\" not not about it to writrtita
or rather far/and the fact holds she possibly farohott\" or or
not look at \"the different kinds here far and mosteand that
somewhat much now better or more power of the User work or just
much more structures, or much its  noWrtnik .\" farll we can
seem the interest of the Thomas specifically not whom clearedch
so far.  Its not or nobody't look or to handle it not he doesnt
stop formerierra  Err he looks.\"::and'and \$ or or not rather
now to pay much more much at=g-- OUT onhis basics,' print and
or \"Yes,'one of''_to know out there\" not remained a+ twisted
or business lot of what Richard would (line truncated to 1000
characters)

SDTT:

by the same.  There is people they can do, that loos a more
In-between much than the has since the after of radio which used
to be an area,\n more low than the pros among a are basketball,
baseball and the, and what of today did they try for more,
but prior to family and college the researchers knew) that the
one that would really be a concerted effort, certainly in the
country is there and.  that in Their recent New Media almost no,
assessedIn Ginn,, in March, an, Ferris brothers found.  everything
from their baby toddlers, did not look good.  There was enough
still for twice.  they published a review in which mentioned
the risk.  and, in the form "The Harris' his still 'm risk was
diminished most The lost lingers following the.il number were\n
vers.  It, they told me the part that they were now prized was the
stiffz type making Ferrier do.  But they wrote out the awesome
Adam Kruger in the factory and, when we talked to, it the, he
were in the studio and they put him many out.  The second D (line
truncated to 1000 characters)

CDLM:

<|endoftext|> had to live on, murder.  The hostages had one row
and the two row of the other.  (The hostage, by Kraft Foods, was
46\n \n with three armed men men in a uniform by policeutives
and certain pop artists, Racist pres Bane11 the song Delta\n
\nsterling heightened the attack aLadderem, said they had\n \n ,
had other hostages thrown as they circled.  Yet the Independent\n

\nnews , and the the raid's days, is much more than when there were\n the possibilities for the same subject.  The Dec said KERS could said, if\nthen AERS did not food New Whorig on board the rear of the Blacks, only so on if the\n \nimmigrants were Indians and the, had most of the Blacks Blacks not be French.  Yet the KERS then asked KERS to claim that kicking a ball for\n \nthe comedy not in the right thing.  The Good of the Brain, the K and Bola Four of the NRL., on the night of the Opening Cere the the Challenge Games, the Mr.  Roberto, who players argued over farmers and pacified movement farmers the the late 70s (line truncated to 1000 characters)

**Step 64:**
MDLM:
<|endoftext|>, the next thing they saw\n \n was a male scream that the victim wanted to kick something,\" Quinn said.  At that point, he got up and started screaming.\"\n \nPolice soon pursued the cab into the bar.\n \n \"They picked up the suspect and left.  They walked to the bathroom and they came to some dried blood coming from the victim's mouth and they located [the suspect inside the bar],\" Quinn said.  \"They're not taking a victim's DNA because obviously, they don't know if the male victim had a knife or both.\"\n \nThe addition was found closed and \"several items that were initially used in the attack, Quinn\n \nSmith said.\n \nThe man still lives near Longleston in Sonoma Monday night and Tuesday March 23.\n \nMcNeil told KING-TV in Southern California, where he met UT A&M student there on Saturday night after a\n \ncar accident there.\n \nThe arrest warrant was processed Tuesday, and police hadn't given much a negative Ephesian police report. \"They remain positive.\"\n \n \"It's (line truncated to 1000 characters)

DUO+DCD:
<|endoftext|> of Posture-17.  Remember amateur domestic monitored product registration stands at 73, and a number of matches have been canceled.\n \nCivil -rights groups have taken part in the struggle against Amnesty and defend the Constitution, the official Global Times reported.\n \nNortino prison executives face questions about arrest\n \nChristian Numel, who have been charged over a July 2012 incident remains behind bars in the Shenyang Dalian-chu jail until Liu in November, the son-in-law of Hui, chief of several members including Prince John Roman's condense Panda conglomerate.  Numel was acquitted on statements made during a Aug.  16 interview by Yong Kushu, a Chinese state radio website.\n \nThe spate of border arrests has signalled the intent of broadening of human rights in China.\n \nThe computers that were captured looking anti-government could be easily labeled as spies by legal experts that they were illegal, and hundreds of names of senior Chinese security officials were (line truncated to 1000 characters)

SDTT:
<|endoftext|> the same thing too.  She also was inspiredShe threw together a group of local kids.  When a lot of kids like Kyle came in to introduce herself to us, it kind of made her realize that she genuinely wanted to go out and be involved in some way.  He was a model type of kid (that's just her imagination:  that's what Los Angeles was for her home).  During her time in St.  Louis, she had already been in some pretty fine little church churches. She was one of those sort of people that I got to surround myself with.\n Yeah, it was an extremely exciting experience.  It was

enough money for you to be behind the camera.\n After that, we did the improv concert, and I didn't want to go and was excited.  I was nervous.  I was flying around and expecting it to sound like it was this kind of a simulator, a real character.  And it was really difficult as a child.  All the cameras are on the stage, and I told my director, "I not have to screw it up.  I just have to do it." Anybody on my money, I just w CDLM:
<|endoftext|> largest union, JPMorgan Chase-DHS Bank, decided to resign earlier this week when the UK voted to liberalise the EU, threatening to pull the country from the customs union.\n \nThe MEP has warned that one of the best ways to leave EU is to leave the Greek exit, which is backed by groups such as the social-democratic Democracy Alliance, which is a member of the Remain campaign.  He warned of further \"political or social risk\" unions being back in the European Union, when a coalition of government removed the backing of its politicians and staff members from its membership.\n \nEarlier this week, the MEP had been resigning from the EU after being alerted to questions over some of his newspaper articles.\n \nThe Dublin-born MEP told a joint news conference in France morning that he was not following the decision to liberalise the EU and use it to prevent the UK from abandoning membership.  This would apparently result in a period of turbulence on the prospects of leaving (line truncated to 1000 characters)

**Step 512:**
MDLM:
<|endoftext|> him when [Singh was about to the] Supreme Court, and said, I hit every benches.  He took a box of 10.  Then he grabbed one and said, Okay, we're going to be OK now.\n \nToward the evening, Ahmadi, the second baseman of the 66, thrown Ram down onto a small concrete ground before spread his wings and clearing the deck for the ceremony.\n \nThe people near the lockers realised immediately that Deepak Ohera, another then policeman, in the eighth over of the sixth innings, had pushed Ram sprayed himself to the ground and fell.\n \nThey ran and chased away all those that did not know what had happened.  Dahla, who had enough power and pace to drive every one and one of the Indians home from college the next day, was killed.\n \n \"They killed me,\" he remembers.  \"It is sad.  I remember it for a minute.  Now, I feel a little more having my family and just going to work hard and play.  I have never felt that.\n \nGusmen table top scoring\n \nOnly 30-year-old 39-year-old Ali Ganes (line truncated to 1000 characters)

DUO+DCD:
"<|endoftext|> after a year and a million days, Sarafaz is about to do the movie, \"Jenna,\" who faces the streets of Virginia.\n \nChristine Nakabub meig:  Judith Kharfatabi did actually see the tape at first, but it was pulled from the November 4 of the initially [main show] screening, now for November 3 because it's really hard to make \"Pappers Out\" to review, like, November 14.  It's \"Noh and Tam\" -- \"It's a Link-in Land.\" And, so I consider it even tougher, four months to make.  It's dissimilar to when I talked with Lois Jordan when she was done.  I said, \"Here's exorcism.\" She said, \"Well, first, Ms.  Out, it was inside.  What does it take to do these sagas in those daysyou'll get your choice.  I'm visiting now with Fox, so it's just kind of certain I knew she's Lady Gaga.  So I'm expecting, it's not

Tammy Jenna.  It's that, indeed.  He's on board, \"computer in the middle of the street,\" at 86 10th Street.  I did Apo Hammer today, on November 15, and this is a great time be (line truncated to 1000 characters)

SDTT:

<|endoftext|> off the floor.  The two didn't happen because we were rested.\n "My son came back and came off to the floor on the opening day out of UCLA, and while I played, I still had my skin covered up from it.  It was a long, tough opportunity to go to play by a great team, but I was kind of a little bit exposed to some of the things that we put him ready to go."\n In the end, I liked that performance against UCLA. He was able to have a very positive positive going through some of the injuries that I've had to deal with.  He went on to have an ankle and ended up leaving the game and actually went down to play.  He was very lively, and we got the win.\n "I've gotten a really good relationship with a lot of my teammates, with the leadership guys in the locker room. With the defense guys, with other guys in the locker room, he was able to play."\n And that worked.  Check out the highlights from the season to come.\n "We haven't recovered from the line or our run or anything like that.  It was fr

CDLM:

<|endoftext|> the EU as the primary focus of civil society, and is fully respected in the EUs role in shaping the global economy.\n \nSo not only is the impact of Polish economy and investment in Poland having on the political and external context of the EU and its investment in many of the European countries, but the reality is the political and external context of the EU is a critical economic partner.  That is why the EU will continue to continue to thoroughly compete with the rest of Europe.\n \nThe EU has one of the worlds largest trading partners for one of the worlds biggest markets and trans-Atlantic European integration.  Though the EU continues to strengthen its position on the EUs global economy, it continues to overcome the challenges that Europeans face from the EU, and its financial commitment through its financial reporting and economic practices.\n \nFrance is one of the largest investments in the EU. In addition to building the international economy to be able to (line truncated to 1000 characters)

