# OpenReview forum: "Consistent Diffusion Language Models"
_ICLR.cc/2026/Conference — Submitted to ICLR 2026_

### Official Review · Reviewer_Cqsk · 2025-10-28

**Soundness:** 3
**Presentation:** 2
**Contribution:** 3
**Rating:** 4
**Confidence:** 4

**Summary:**

This paper proposes a novel family of generative models for fast inference in diffusion LLMs, known as consistency diffusion language model (CDLM), through enforcing multi-path consistency of the denoiser model. The model is trained by the combination of a standard cross-entropy loss and a consistency loss for self-distillation, and during inference, it generates in the style of continuous consistency models. The authors discussed the design space of CDLM and evaluated it on various language modeling tasks.

**Strengths:**

The paper tackles an important issue of diffusion LLMs, which is the slow inference speed caused by many denoising steps. The proposed CDLM is an interesting approach to address this from the modeling perspective, and from the reported results, it achieves SOTA performance compared with other model structures, such as the vanilla MDLM, SDTT and DUO. Moreover, the loss does not require distillation from a teacher model (i.e., can be trained from scratch), which is a nice property. The authors also provide conceptual comparisons with related methods such as the vanilla masked diffusion model, consistency models, and distillation-based methods. The experimental results are comprehensive, covering various ablation studies.

**Weaknesses:**

The main weakness of the paper is the confusing writing and presentation. Also, I don't fully understand the motivation and feel that the learning objective is not well defined. I won't lean towards acceptance unless these issues are sufficiently addressed in the rebuttal.

1. The notations in the paper are mainly **token level** instead of **sequence level**. For instance, in section 2, $\mathbf{x} _ 0\in\mathcal{X}$ is defined as a *sequence* of one-hot tokens (so I thought $\mathcal{X}\subset(\\{0,1\\}^V)^{|\mathbf{x}|}$), but the transition matrix $\mathbf{Q} _ t$ has dimensions of $V\times V$, so I realize $\mathbf{x} _ 0$ still represents a single token and $\mathcal{X}\subset\\{0,1\\}^{V}$. Everything after that (except in algorithm 2) seems to be all operating on token level. However, we all know that the actual language modeling task is on sequence level. While I understand this simplifies the presentation to some extent, it loses many important implementation details for dealing with sequences, and made me very confused when reading the paper. I hope the authors can clarify this point. I also suggest removing the one-hot representation and using token indices instead, as otherwise the matrix multiplications with $\mathbf{Q} _ t$ would be hard to follow.

2. The learning objective is also very unclear to me due to the token-level notation. First, in definitions 2 and 3, what's the range of $f$ when we input a partially masked sequence $\mathbf{x} _ t$ and a time $t$? Is it $(\Delta^V)^{|\mathbf{x} _ t|}$ or something else? If it is $(\Delta^V)^{|\mathbf{x} _ t|}$, (i) is there any mechanism to model the correlation between different masked positions, as we are doing few-step generation that would involve unmasking multiple positions simultaneously, (ii) for non-mask dimensions in $\mathbf{x} _ t$, will the output at those dimensions matter (because in the cross-entropy loss for training masked diffusion models, only the logit output at masked positions are used to compute the loss), and (iii) what's the explicit form of the loss $\mathbb{D}(f _ \theta(\mathbf{x} _ t,t)||f _ {\bar\theta}(\mathbf{x} _ s,s))$ in equation (9), considering the dimensions that are masked in $\mathbf{x} _ t$ but unmasked in $\mathbf{x} _ s$? I think clarifying these points will help readers better understand the learning objective. Also, it would be helpful if the authors could provide the code in the link to the anonymous repository so that we can better understand the implementation details, which is currently empty.

3. I feel equation (8) is not well-defined. The optimal predictor $f ^ * $ is defined to satisfy the equality $f ^ * (\mathbf{x} _ t,t)=\mathbb{E} _ {q(\mathbf{x} _ s|\mathbf{x} _ t,\mathbf{x} _ 0)}[f ^ * (\mathbf{x} _ s,s)]$. The left-hand side does not depend on $\mathbf{x} _ 0$, while the right-hand side does. There are infinitely many clean data sequences $\mathbf{x} _ 0$ that can be masked into $\mathbf{x} _ t$, so how can the equality hold for all possible $\mathbf{x} _ 0$? Judging from lemma 2, it seems that we need to take an expectation over $\mathbf{x} _ 0$ on the right-hand side as well, i.e., assume that I can swap the order of $\mathbb{E}$ and $\mathbb{D}$, we want to match $f ^ * (\mathbf{x} _ t,t)$ with $\mathbb{E} _ {q(\mathbf{x} _ 0|\mathbf{x} _ t)}[\mathbb{E} _ {q(\mathbf{x} _ s|\mathbf{x} _ t,\mathbf{x} _ 0)}[f ^ * (\mathbf{x} _ s,s)]]=\mathbb{E} _ {q(\mathbf{x} _ s|\mathbf{x} _ t)}[f ^ * (\mathbf{x} _ s,s)]$ as the dependency on $\mathbf{x} _ 0$ is marginalized out. I think this is a more reasonable definition of the optimal predictor, but whether this swap is applicable is unknown, so I am not sure if this is what the authors intended. Finally, does the optimal predictor have an analytical expression if we take $\mathbb{D}$ to be some simple divergence such as KL or $L^2$? Of course, if the swap is valid, i.e., $f ^ * (\mathbf{x} _ t,t)=\mathbb{E} _ {q(\mathbf{x} _ s|\mathbf{x} _ t)}[f ^ * (\mathbf{x} _ s,s)]$, we know it is the posterior distribution $\\{q(\mathbf{x} _ 0|\mathbf{x} _ t):\forall\mathbf{x} _ 0\\}$.

**Questions:**

My main questions are detailed in the weaknesses above. In addition, I also have a few more minor comments:

1. The presentation of figure 2 can be improved to be more clear and have better visual effects.

2. I don't think it's necessary to present the transition matrix $\mathbf{Q} _ t$. For masked diffusion models, the forward and backward probabilities as well as the posterior bridge operator can be defined directly.

3. From equation (10), is it possible to finetune (distill) a pretrained masked diffusion model into a CDLM? This may be an interesting experiment to try.

4. Assume $\mathbb{D}$ is symmetric, is there any intuitive reason for preferring $\mathbb{D}(f _ \theta(\mathbf{x} _ t,t)||f _ {\bar\theta}(\mathbf{x} _ s,s))$ over $\mathbb{D}(f _ {\bar\theta}(\mathbf{x} _ t,t)||f _ \theta(\mathbf{x} _ s,s))$?

5. When sampling from CDLM, the procedure described in section 3.1 can't correct the generated tokens at previous steps, as once a token is generated it will always be fixed under the posterior bridge operator. This is a little bit different from the continuous consistency model where the predicted $\mathbf{x} _ 0$ will be renoised so that all dimensions are updated. Do you think including a correction mechanism would help improve the performance?

---

> ### Author Response · Authors · 2025-12-03
>
> We are thankful for the detailed, constructive feedback and for highlighting our SOTA performance and comprehensive ablations. We apologize for the confusion regarding notation and are happy to provide concrete clarifications below.
>
> **1. Notation and Learning Objective: Sequence vs. Token**
>
> You correctly pointed out that the paper can be more cautious about mixing token-level math with sequence-level implementation. We agree this needs to be explicit and have included the following clarification:
>
> * **Sequence Definition:** $x_t \in \mathcal{V}^L$ is a sequence of length $L$.
> * **Factorized Corruption:** The corruption process factorizes per-token: $q(x_t|x_0) = \prod_{i=1}^L q(x_{t,i}|x_{0,i})$.
> * **Model Output:** The model $f_\theta$ is a Transformer that processes the full sequence to capture correlations (answering your question (i)), producing per-position distributions $f_{\theta,i}(x_t, t) \in \Delta(\mathcal{V})$.
>
> **Explicit Loss Form (answering ii & iii):**
> The consistency loss is computed only on positions masked at time $t$. If a position $i$ is masked at time $t$ but unmasked at time $s$ (due to the bridge step), the unmasked value at $s$ serves as the target signal. We have updated Section 3 to use this explicit sequence-level formulation.
>
> **2. Theoretical Soundness: Eq. 8 and $x_0$**
>
> You asked how Eq. 8 holds if the RHS depends on $x_0$. This is a crucial theoretical point.
>
> * **The Bridge:** The Posterior Bridge defines the "valid" path from $x_t$ back to a *specific* $x_0$. Without $x_0$, the bridge is undefined.
> * **Marginalization:** You correctly intuited that we must marginalize out $x_0$. **This is exactly what the CDLM objective does.** By minimizing the expected divergence $\mathbb{E}\_{x_0, x_t}[\dots]$, $f_\theta(x_t)$ learns to predict the *expected target*, effectively learning the posterior marginal $p(x_0 | x_t)$.
> * **Revision:** We have rewritten Eq. 8 to explicitly include the expectation over the data distribution, removing the confusing dependence on a specific $x_0$.
>
> **3. Design Choices: Divergence and Sampling**
>
> **Why JSD?** Empirically, Forward KL tends to be mode-seeking (leading to collapse), while Reverse KL tends to be mode-covering (leading to poor sample quality). JSD is symmetric and bounded, providing a stable middle ground that enables training from scratch without an external teacher.
>
> **Correction Mechanism:** You correctly note that the sampler does not "correct" past tokens, due to the absorbing state prior (once a token is unmasked, it stays unmasked). However, the CDLM *training* forces the *predictor* $f_\theta$ to update its belief about the *entire* sequence at every step. We agree that adding a predictor-corrector mechanism (like ReMDM) is a promising extension for future work.
>
> **4. Additional Questions**
>
> **Finetuning MDLM:** Yes, Eq. 10 suggests that finetuning a pre-trained MDLM into a CDLM is possible by gradually introducing the consistency loss. We focused on training from scratch to demonstrate the method's standalone capability, but finetuning is a valid use case.
>
> **Presentation:** The general transition matrix definitions emphasize the generality of our work. We focus on masked diffusion but our framework (verifiably) extends to other forms of diffusion like uniform diffusion.

---

### Official Review · Reviewer_bf5n · 2025-10-30

**Soundness:** 3
**Presentation:** 2
**Contribution:** 3
**Rating:** 2
**Confidence:** 4

**Summary:**

This paper proposes the Consistent Diffusion Language Model (CDLM), a new type of diffusion language model resembling Consistency Models in the continuous domain. At its core is the observation that the ground-truth posterior between two random time steps $s$ and $t$ is analytically tractable, and thus we can leverage this to perform consistency training, with accelerated model inference as a natural outcome. The authors demonstrated that, when combined with the CDLM training loss, MDLM models yield better generation quality than standard models like MDLM and DUO, and that CDLM is also amenable to SDTT distillation. The authors observed that CDLM+SDTT further improves generation quality and that CDLM+SDTT outperforms MDLM+SDTT and DUO+DCD.

**Strengths:**

* The high-level idea is intuitive and straightforward, making the paper easy to follow.
* The storyline of the paper is good.
* The authors conducted a comprehensive analysis, both theoretically and empirically.
* It is apparent that the authors put forward many engineering tricks that are crucial to the success of CDLM. I think these tricks are very valuable to the community.

**Weaknesses:**

* The mathematical notation system is not rigorous enough for a top-tier machine learning paper. In Section 2, $\mathbf{x}$ is explicitly stated to represent a token sequence. Suppose the token sequence has $L$ tokens, then $\mathbf{x}$ should have the shape of $L \times |V|$, and the $\mathbf{Q}$ matrices should have the shape of $|V| \times |V|$. If this is correct, then in equation 6, the term $\mathbf{Q}\_{s+1:t}^T\mathbf{x}_t$ does not make sense. I believe there is either a typo or some fundamental issues, and I strongly advise the authors to do a double-check.
* One of the core results of the paper is that CDLM training is better than MDLM alone, but there is a fundamental problem. Because the training loss contains another forward pass on the $\mathbf{x}_s$ sequence as well, when the training steps are the same, the training FLOPs double. Hence, I do not believe the authors have conducted a fair comparison in their paper. I suggest the authors present results with comparable FLOPs to make the comparison fair.
* It is unclear how well the model is trained since CDLM is not an inference-time only method. Crucial results, such as validation perplexity and zero-shot perplexity across different datasets, are missing.
* Both SDTT and DUO presented results with cascaded distillation, i.e., several rounds of distillation, but in this paper, the authors only conducted experiments using one round of distillation, making the results less convincing.
* The writing of the paper seems overly verbose and not logical. Definitions 2 and 3 are neither used in the algorithm nor in any of the subsequent theoretical analyses. Section 3.3 also seems overly verbose for the method part. A more succinct and mathematical presentation should be expected.

**Questions:**

* As mentioned in the weakness part, how do the training dynamics evolve for CDLM and MDLM when the same computational budget is used?
* What are the validation perplexity numbers and zero-shot perplexity numbers of CDLM and MDLM on different datasets?
* How does the generation quality evolve for CDLM+SDTT, MDLM+SDTT, and DUO+DCD when multiple rounds of distillation is conducted?
* Can the authors show any evidence that CDLM is better than existing methods in any of the data modalities other than natural language?
* How does CDLM perform on downstream tasks?
* How does CDLM perform against predictor-corrector samplers such as ReMDM [1] in the domain of fast generation?

---
**References**
1. Wang, Guanghan, Yair Schiff, Subham Sekhar Sahoo, and Volodymyr Kuleshov. "Remasking discrete diffusion models with inference-time scaling." arXiv preprint arXiv:2503.00307 (2025).

---

> ### Author Response · Authors · 2025-12-03
>
> We are thankful for the rigorous review. We are glad you found the high-level idea intuitive, the storyline good, and the engineering design choices valuable. We address your main concerns regarding mathematical rigor and computational fairness below.
>
> **1. Mathematical Rigor: Token vs. Sequence Notation**
>
> You correctly pointed out that the current notation does not cautiously handle sequence-level and token-level views, leading to potential confusion about matrix shapes in Eq. 6.
>
> * **Clarification:** Our formalism follows the standard convention in discrete diffusion literature (e.g., D3PM, Austin et al., 2021a), where the transition matrices $Q_t \in \mathbb{R}^{|\mathcal{V}|\times|\mathcal{V}|}$ and the corruption process are defined **factor-wise per token**, while the model $f_\theta$ processes the full sequence $x_t \in \mathcal{V}^L$ to capture dependencies.
> * **Revision:** We agree this should be explicit. We have rewritten Section 2 to clearly distinguish the per-token corruption process ($q(x_t|x_0) = \prod q(x_{t,i}|x_{0,i})$) from the sequence-level model architecture. We have also streamlined the theoretical exposition by handling definitions 2 and 3 better.
>
> **2. The "Double FLOPs" Critique**
>
> You noted that CDLM involves two forward passes per step (to generate $x_s$ and $x_t$), raising an evaluation fairness concern against MDLM. This is a critical point. While the actual overhead is less than $2\times$ (as the target network requires no backward pass), we can work with the premise of "double compute" and ran a new baseline experiment to address this head-on.
>
> **New Experiment:** We trained the baseline MDLM for double the steps than our CDLM budget to ensure the baseline utilized *more* total FLOPs than our method.
> Even when the baseline is given 2x the training steps, it still falls far short of the CDLM performance. This helps establish that our gains stem from the algorithmic superiority of CDLM, not merely additional compute. Furthermore, CDLM reduces *inference-time* costs by 10-100$\times$, which is the primary bottleneck for deployment.
>
> **3. Distillation Rounds and Validation Metrics**
>
> Our SDTT implementation utilizes a continuous update schedule over 50k steps, which is effectively equivalent to **5 rounds** of the standard 10k-step protocol used in DUO/DCD. Thus, our comparison is matched in "distillation depth." We will clarify this in the text.
>
> We agree that validation perplexity can be useful, yet our focus is on generative capabilities best captured by generative perplexity that we report (similar to say FID being important on image data). We did not include validation perplexity numbers due to concern of them being misinterpreted.
>
> **4. Generalization and Baselines**
>
> **ReMDM:** We view ReMDM [1] as a complementary *inference-time* sampler, whereas CDLM is a *training-time* objective. As such, they are not mutually exclusive and one could likely apply ReMDM-style correctors on top of a CDLM base. We can enhance the discussion on this topic (**L462**) to state how inference techniques like ReMDM can potentially improve CDLM further.
>
> **Other Modalities:** While the title emphasizes *Language* Modeling, the multi-path consistency framework applies to any discrete data with a Markov corruption process (e.g., DNA, Code). We focus on language here to provide a comprehensive evaluation against strong baselines. We have still added an experiment with a different form of diffusion (UDLM), and find our proposal to help there too.

---

### Official Review · Reviewer_jXUn · 2025-11-01

**Soundness:** 3
**Presentation:** 3
**Contribution:** 2
**Rating:** 4
**Confidence:** 4

**Summary:**

The paper proposes CDLM, a consistency training algorithm for training a few-step masked diffusion model from scratch. CDLM considers consistency distillation-type objectives without leveraging concepts like PFODE and combines the vanilla MDM training loss to enable training a few-step discrete generative model from scratch. Experiments on GPT-2-level text generation tasks show promising performance.

**Strengths:**

- The paper considers an important problem of training a few-step discrete generative problem, and provides a novel solution to it.
- CDLM is one of the first successful consistency training algorithms for masked diffusion models applied to text generation.
- The demonstrated empirical performance is compelling, and outperforms most of the existing baselines.

**Weaknesses:**

- While motivations to derive the training loss are valid, the technical derivations are flawed. I am not convinced that Eq. 8 admits a unique or well-defined solution.  This essentially states that there exists an x0 prediction map f that behaves like a "flow map" for the MDM. Imagine x_t is a sequence full of masked tokens. How does f^* even exist in this case?
- The distillation loss highly resembles the consistency loss for the continuous diffusion model, while there are no deterministic trajectories provided to ensure the existence of validity of the consistency loss. This makes me wonder about the theoretical guarantee of the method and if it is ready to scale to a larger size.
- Some important experimental details are missing. For example, the paper does not discuss what CDLM-PPL Optimized is while still listing its performance in the table. Besides, optimized for ppl but loses diversity (entropy) makes me feel worried that the trained model will produce meaningless, low perplexity text frequently.
- The given anonymous GitHub repo is empty, and therefore, I can't verify the technical implementation.

**Questions:**

See weakness for most of the questions. The following are some additional ones.
- In Table 1,  CDLM–PPLOptimized produces better ppl than AR model. How is CDLM-PPLOptimized implemented and how is this possible?
- In Table 1, CDLM-SDTT in general has the lowest entropy across all distilled methods. Why is that? And also, as a few-step generation model, why is progressive distillation with SDTT necessary?
- In Table 2, why is only DUO with the greedy sampler listed, while DUO itself is not? The low diversity of DUO results seems to be related to the greedy sampler.

---

> ### Author Response · Authors · 2025-12-03
>
> Thank you for the positive assessment of our empirical performance and for recognizing CDLM as one of the first successful consistency training algorithms for masked diffusion LMs. We are happy to clarify the theoretical validity and experimental details below.
>
> **1. Theoretical Soundness: The Existence of $f^*$ and Deterministic Trajectories**
>
> You correctly pointed out that Eq. 8, as written, implies an ill-posed equality depending on $x_0$, and you raised concerns about the lack of deterministic trajectories. We agree that the current presentation is informal, but the underlying theory is sound.
>
> **The Optimal Predictor ($f^*$):** The intended definition is that $f^*(x_t, t)$ targets the **posterior marginal distribution** $p(x_0 \mid x_t)$, not a deterministic point estimate. This is well-defined even when $x_t$ is fully masked (where it equals the unconditional dataset marginal $p(x_0)$). The consistency operator $\mathcal{C}\_{s\leftarrow t}$ enforces that the predictor at time $t$ matches the *expected* prediction at time $s$ (drawn from the posterior bridge). By the Markov property, $p(x\_0 \mid x\_t) = \mathbb{E}\_{x\_s \sim q(x\_s \mid x\_t)}[p(x\_0 \mid x\_s)]$, making the posterior marginal the fixed point. In revised manuscript, we explicitly include the expectation over the data distribution to remove the ambiguity.
>
> **No Deterministic Trajectories Needed:** We fully agree that discrete diffusion lacks the deterministic PF-ODE trajectories found in continuous models. This absence is precisely why we propose multi-path consistency objective—which we have rebranded as Multi-Path **Discrete** Consistency to make its distinction from continuous consistency models even more explicit—which enforces path-invariance *in expectation* over stochastic bridges rather than along a single ODE path. Our theoretical guarantee is that minimizing this objective under a proper scoring rule recovers the Bayes-optimal posterior $p(x\_0 | x\_t)$, ensuring stability without requiring deterministic trajectories.
>
> **2. Experimental Details: CDLM-PPLOptimized and Diversity**
>
> You raised concerns about the definition of CDLM-PPLOptimized and its potential for "meaningless low perplexity."
>
> **Implementation Details:** We apologize if this was easy to miss, but the details for CDLM-PPLOptimized are provided in the paper. With training details provided in **S4.2 (L350-356)**, it is not a separate architecture but a variant using a modified schedule (gradually shrinking $\Delta_T$) in the final 50k steps. This allows us to trade some diversity for fidelity, effectively moving along the quality-diversity frontier.
>
> **Avoiding Collapse:** While this CDLM-PPLOptimized has lower entropy than the other variant, it is still quite high and does not collapse to meaningless text (where entropy is often $<3$). Its ability to outperform AR models in generative perplexity (Table 1) suggests that for $>32$ steps, the iterative refinement of diffusion (correcting early errors) surpasses the error accumulation of greedy AR decoding.
>
> **3. Comparisons with SDTT and DUO**
>
> **Distillation Rounds:** You asked why we include SDTT and noted its low entropy. We include SDTT to demonstrate that CDLM is a superior **base model**. Using SDTT is not "necessary" but complementary, and we run these additional experiments to put this point across. Our implementation uses a schedule equivalent to 5 distillation stages (50k steps), matching the protocol in standard baselines (like DUO). The low entropy of CDLM+SDTT indicates it is an extremely effective "mode-seeker," sharpening the distribution more effectively than MDLM+SDTT while avoiding collapse.
>
> **DUO Baseline:** We focused on DUO-greedy as a generally stronger baseline. We can add standard DUO results too for completeness.
>
> **4. Code Availability**
>
> We will upload a minimal reference implementation at the initially shared link to allow verification of the technical components. Full code and trained models will be released upon acceptance.

---

### Official Review · Reviewer_Jjmk · 2025-11-02

**Soundness:** 3
**Presentation:** 2
**Contribution:** 2
**Rating:** 4
**Confidence:** 4

**Summary:**

This paper introduces the "Consistent Diffusion Language Model" (CDLM), which is trained with a new training objective that enforces consistency between denoising predictions at some state $x_t$ and denoising predictions at less noisy states $x_s$ that are drawn from the posterior $q(x_s|x_t, x_0)$. Training discrete diffusion models like this is inspired by consistency models from the continuous diffusion literature, and the authors find that CDLM can achieve improved results when generating with few sampling steps, compared to relevant baselines. The experiments consist of unconditional and conditional text generation benchmarks, measuring generative perplexity and entropy. The paper also includes ablation studies.

**Strengths:**

**Results:** CDLM achieves improved results (as measured in generative perplexity) when generating with few sampling steps compared to baselines.

**Originality:** The proposed method is novel, to the best of my knowledge (although it is closely related to Self-Distillation Through Time (SDTT)).

**Significance:** Accelerating diffusion language models is an important research direction that is currently getting a lot of attention, and the paper addresses this problem. Therefore, the work can be considered significant.

**Weaknesses:**

**Presentation and relation to regular Consistency Models:** The paper motivates CDLM from the perspective of consistency models from the continuous diffusion literature, even calling their method "*Consistent* Diffusion Language Model". I think this is misleading, because the method really behaves very differently compared to regular consistency models. Since the posterior bridge involves an expectation over $x_s$, training CDLM necessarily always incorporates some form of averaging over different $x_s$ and the corresponding predictions, which prevents learning a perfect few-step denoiser. This is in contrast to the ODE-based consistency objective in regular consistency models, which does not incorporate such an expectation. For instance, I believe it is fundamentally impossible to learn an accurate one-step denoiser from all-mask states with CDLM's objective, while this is possible in regular consistency models. Hence, the way the work is presented is misleading, as readers may falsely get the impression that this is a direct generalization of consistency models to diffusion language models, but this is not the case. I would suggest the authors to better discuss this and present the method accordingly.

**Theoretical justification:** Related to the above point, the paper misses a rigorous theoretical justification for the proposed objective. For regular consistency models, as discussed above, the perfect minimizer of the objective would be a perfect one-step generator, maintaining the original generated distribution. What would the perfect minimizer of the proposed objective here correspond to, given the averaging over $x_s$? Again, this is a very different situation compared to regular consistency models.

**1-step results:** CDLM seems to completely fail in single-step generation (Figure 1). This is again evidence that the model behaves entirely different compared to regular consistency models, which can produce good single-step samples.

**Relation to SDTT:** The method is similar to Self-Distillation Through Time, with the main difference being that here we are training from scratch, whereas SDTT uses a teacher.

**Ad-how modifications and hyperparameters:** It seems the method can only be trained to high performance with additional modifications (mixing in a regular diffusion language model loss) and carefully chosen hyperparameters, like step size $\delta$ and $\kappa_{ms}$ schedules. The necessity for these adaptations calls into question how principled the approach is, aligned with the comments above.

**Quality-diversity tradeoff:** The numerical results suggest that there is a quality-diversity tradeoff, also evidenced by the need to train two different models (CDLM vs. CDLM-PPLOptimized). While the method achieves strong generative perplexity, this seems to come with reduced entropy. The reason for this is not discussed in detail, and -- related to my above points -- this is also fundamentally different from regular consistency models, which learn the exact noise-to-data mapping without loss in diversity. This again shows that the proposed method is entirely different from consistency models.

**Questions:**

The paper says it uses the $w(\delta)=1/\delta$ weighting *to help path length normalization*. Can the authors explain this better? According to my understanding $\delta$ essentially corresponds to the path length in time. If we wanted *each unit of "time" on the corruption axis contribute equally* (quoting from line 268) then why use the *inverse* $\delta$, i.e. $1/\delta$ as weighting, and not $\delta$ directly? The current formulation gives more weight to short paths (small $\delta$) and less to longer ones. It would be great if the authors could explain this.

---

> ### Author Response · Authors · 2025-12-03
>
> We sincerely thank you for the thoughtful review and for recognizing the novelty, significance, and strong results of our work. You correctly point out the distinction between CDLMs and continuous consistency models (CMs), which forms the conceptual core of our work, and we accept that our presentation could make this clearer.
>
> **1. Relation to Consistency Models: Not a Mismatch but the Core Contribution**
>
> We are fully on board with your insight that CDLM differs fundamentally from continuous CMs. Indeed, we view this distinction not as a weakness or misleading presentation, but as the **primary conceptual contribution** of our work.
>
> As identified by you, as well as briefly discussed in the paper (**L180-188**), continuous CMs rely on a unique, deterministic PF-ODE trajectory, where "consistency" implies invariance along this single path. In the discrete reality, no such ODE exists. Therefore, we cannot simply "port" consistency models to discrete diffusion setting by somehow "discretizing" the consistency notion from continuous CMs (which prior literature including DUO has struggled with).
> Instead of attempting to discretize a non-existent ODE, our work proposes a new notion of "multi-path consistency". This *generalizes* the consistency principle to the *stochastic discrete setting* by enforcing path-independence *in expectation* over the family of exact stochastic posterior bridges $q(x_s \mid x_t, x_0)$. To distinguish from continuous CMs, we emphasize the notion of *multi-path* consistency and use slightly different naming of *Consistent* (instead of Consisten*cy*) Diffusion Language Models. To emphasize the distinction even further, we can rename multi-path consistency to **"multi-path discrete consistency"**.
>
> **Theoretical Justification and the "Perfect Minimizer":**
> This stochastic formulation directly answers your crucial question regarding the "perfect minimizer." We define the Bayes-optimal predictor as $f^*(x_t, t) := p(x_0 \mid x_t)$ (the posterior marginal). By the Markov property of the forward process, we can derive:
>
> $$
> p(x_0 \mid x_t) = \mathbb{E}_{x_s \sim q(x_s \mid x_t)}[p(x_0 \mid x_s)].
> $$
>
> Thus, $f^*$ is a fixed point of the operator. Under a strictly proper scoring rule (like the JSD we employ), the *unique minimizer* of our objective is *not* a deterministic flow map but exactly the **posterior distribution over clean data** $p(x_0 \mid x_t)$.
>
> **Why 1-step Generation "fails" (and why this is expected):**
> This theoretical insight explains the 1-step behavior observed in Figure 1. In masked discrete diffusion, when $x_t$ is fully masked, the optimal predictor $p(x_0 \mid x_t)$ is simply the unconditional data distribution. Because this distribution is highly *multimodal*, a single-step sample (without autoregression) yields independent tokens lacking global coherence. This is a fundamental property of the discrete masked setting, not a flaw in the objective. CDLM is explicitly designed for the **few-step regime**, where it successfully builds coherent structure and significantly outperforms baselines that require 100+ steps.
>
> **2. Distinctions from Self-Distillation Through Time (SDTT)**
>
> While related, CDLM differs from SDTT in fundamental ways. First, as you correctly point out, CDLM is a **teacher-free** method "trained" from scratch, whereas SDTT relies on "distilling" a pre-trained MDLM teacher. Second, CDLM utilizes **exact analytic posterior bridges** between any pair of timesteps, whereas SDTT relies on approximate transitions via teacher rollouts. Finally, our method enforces **global multi-path invariance**, whereas SDTT enforces consistency along a specific local schedule.
>
> Additionally, our framework is general enough to apply beyond masked diffusion. We confirm this with our **new results** on Uniform Diffusion, demonstrating that CDLM works for both Uniform and Masked priors, whereas SDTT was demonstrated primarily on MDLM.
>
> Moreover, SDTT which optimizes for few-step generation is actually **complementary** to CDLM which obtains effective few-step generation as a side effect. We even run experiments (Table 1) showing how applying SDTT on top of a CDLM base (CDLM+SDTT) yields the strongest performance.

---

> > ### Author Response · Authors · 2025-12-03
> >
> > **3. Principled Design Choices and Quality-Diversity Tradeoff**
> >
> > While choices like the anchor loss and JSD can seem "ad-hoc," we view these as **principled regularizers** necessary to resolve an under-determined fixed-point problem. Pure self-consistency admits degenerate solutions (e.g., uniform predictions); the max-step anchor loss ties the predictor to the true data distribution at $s=0$, selecting the Bayes-optimal fixed point. Similarly, JSD balances the mode-seeking behavior of forward KL and the mode-covering behavior of reverse KL, preventing collapse. We elaborate on these "design insights" in **S3.2** and complement them with ablation studies (**S4.4 and appendix**).
> >
> > Regarding the tradeoff, we clarify that the base CDLM achieves substantial PPL gains while maintaining high entropy comparable to MDLM. The CDLM-PPLOptimized variant is an optional configuration (analogous to lowering sampling temperature) that moves along the quality-diversity frontier, which has been observed in prior work too (e.g., DUO vs DUO-Greedy).
> >
> > **4. Path Length Normalization ($w(\delta) = 1/\delta$)**
> >
> > The intuition for the inverse weighting is path length *normalization*. A consistency constraint between $t$ and $s=t-\delta$ covers a segment of length $\delta$. If $\delta$ is sampled uniformly, longer segments are rarer but cover more "time volume," while shorter segments are frequent but cover less. Weighting by $1/\delta$ ensures that, in expectation, each unit of "corruption time" contributes equally to the total loss, balancing the learning signal across short and long paths. Alternately, the $1/\delta$ schedule could also be interpreted through a curriculum learning lens: longer paths are inherently difficult and we do not want to strongly penalize the model for not learning what is inherently vague (due to larger number of final possibilities). Nonetheless, we still view specific weighting function as a design choice and alternate, careful choices could improve results further.

---

### Author Response · Authors · 2025-12-03
**Note to AC: Post-Rebuttal Summary**

We are thankful to the reviewers for their constructive feedback and for recognizing the work's novelty and empirical strength. While the initial scores were conservative, we found these to stem primarily from two valid, and easily fixable, misunderstandings: the relationship to continuous consistency models and notational ambiguity. We thoroughly addressed both in individual response to each reviewer, while also strengthening the value of our proposal through additional experiments for (consistent) uniform diffusion as well as FLOPs-matched reporting. We are confident that these clarifications resolve the root causes of the initial ratings, and would have warranted a significant score increase had the process permitted updates.

**1. The Key 'Weakness' is the Core Contribution: Multi-Path Discrete Consistency**

Reviewers Jjmk and jXUn questioned the lack of a deterministic Probability Flow ODE (PF-ODE), which underpins continuous consistency models.

* **Rebuttal:** We clarify that the *absence* of an ODE in discrete space is not a flaw, but the **fundamental motivation** for our work. We were fully on the same page as the reviewers that we cannot simply discretize an ODE that doesn't exist. This challenge is explicitly acknowledged in prior work like DUO (Sahoo et al., 2025), which resorts to constructing a "Deterministic Discrete Trajectory" by projecting a Gaussian ODE onto discrete space as a proxy. We have clarified how CDLM is *not* a discretized version of continuous consistency models but a more significant conceptual extension.
* **Innovation:** In contrast to a specific (Gaussian) proxy like DUO, **Multi-Path Discrete Consistency** offers a general, native solution. We generalize consistency to the *stochastic* regime by enforcing path-invariance *in expectation* over the family of exact stochastic posterior bridges.
* **Theoretical Rigor:** We demonstrated that the unique minimizer of our objective is the Bayes-optimal posterior $p(x_0|x_t)$, offering a distributional analogue of consistency in continuous space and providing a solid theoretical foundation within discrete diffusion literature.

**2. Resolving Presentation Concerns**

Reviewers bf5n and Cqsk found the notation mixing sequence-level models with token-level corruption confusing.

* **Resolution:** This formalism is standard in seminal discrete diffusion works (e.g., D3PM, Austin et al. 2021). To eliminate ambiguity, we have explicitly distinguished sequence-level variables from token-level factors in the revision and rewritten Eq. 8 to include the expectation over data.

**3. State-of-the-Art Performance & Robustness**

All reviewers acknowledged the strong empirical results, where CDLM achieves SOTA generative perplexity in the few-step regime.

* **Compute Fairness:** Reviewer bf5n noted that CDLM uses $\sim 2\times$ compute per step. To prove our gains are algorithmic, we ran a new experiment training the MDLM baseline for $2\times$ our budget, and found that CDLM still massively outperforms the compute-enhanced MDLM.
* **Generalization (New C-UDLM Results):** We even confirmed that our findings extend beyond masked priors by training a Consistent Uniform Diffusion Language Model (C-UDLM). C-UDLM was a massive improvement over standard UDLM, and even offered significant improvements over DUO+DCD (which is a uniform diffusion model incorporating consistency distillation of sorts). These results help demonstrate that our framework is generic and applicable to multiple diffusion mechanisms, unlike prior methods tied to specific priors.


We believe CDLM represents a necessary and principled leap for discrete diffusion, moving beyond the "ODE-discretization" mindset to a truly native discrete formulation. Given that the critiques focused on presentation and a theoretical formulation (stochastic consistency) that we have now clarified, and considering the demonstrated generalizability and SOTA performance, we request the AC to kindly consider the significance of this contribution for acceptance. We are confident that with these clarifications, the work merits a strong positive evaluation.

---

### Meta-Review · Area_Chair_ZiAq · 2026-01-07

**Summary:**

This paper proposes Consistent Diffusion Language Models (CDLM), introducing a multi-path consistency objective for discrete diffusion models aimed at enabling high-quality few-step generation. Reviewers agreed that the paper addresses an important problem and reported strong empirical results, particularly in low-step regimes. However, the initial reviews raised significant concerns regarding theoretical clarity, correctness and rigor of the proposed objective, presentation and notation, computational fairness of comparisons, and interpretation of quality–diversity trade-offs. While the rebuttal provided detailed clarifications, additional experiments, and revised explanations, these changes were not sufficient to fully resolve the core concerns at the level expected for acceptance.

**Reviewer Concerns:**

Reviewers expressed substantial concerns about the conceptual framing and theoretical foundations of the method, particularly the absence of deterministic trajectories in discrete diffusion and whether the proposed consistency objective admits a well-defined and meaningful solution comparable to continuous consistency models. Multiple reviewers found the presentation confusing due to mixing token-level and sequence-level notation, leading to ambiguity in the learning objective and mathematical expressions. There were also concerns about fairness of training comparisons given the additional forward passes required by CDLM, missing validation and zero-shot metrics, entropy reduction and quality–diversity trade-offs, and incomplete or unclear baseline comparisons. Although the rebuttal clarified the intended interpretation of multi-path discrete consistency, revised the mathematical formulation, and added compute-matched experiments and further discussion, key concerns about soundness, clarity, and evaluation rigor remain only partially addressed.

**Reviewer Scores:**

Reviewer scores initially centered around marginally below the acceptance threshold, with at least one clear reject. After the rebuttal, the score distribution would remain mixed, with reservations about correctness, clarity, and evaluation methodology. Overall, the post-rebuttal signal does not clearly shift the paper above the acceptance threshold.

---

### Decision · Program_Chairs · 2026-01-26

Reject